# RestoreVAR: Visual Autoregressive Generation for All-in-One Image Restoration

**Sudarshan Rajagopalan, Kartik Narayan & Vishal M. Patel**
Johns Hopkins University
Baltimore, MD 21218, USA
`{sambasa2,knaraya4,vpatel36}@jhu.edu`

## Abstract

The use of latent diffusion models (LDMs) such as Stable Diffusion has significantly improved the perceptual quality of All-in-One image Restoration (AiOR) methods, while also enhancing their generalization capabilities. However, these LDM-based frameworks suffer from slow inference due to their iterative denoising process, rendering them impractical for time-sensitive applications. Visual autoregressive modeling (VAR), a recently introduced approach for image generation, performs scale-space autoregression and achieves comparable performance to that of state-of-the-art diffusion transformers with drastically reduced computational costs. Moreover, our analysis reveals that coarse scales in VAR primarily capture degradations while finer scales encode scene detail, simplifying the restoration process. Motivated by this, we propose RestoreVAR, a novel VAR-based generative approach for AiOR that significantly outperforms LDM-based models in restoration performance while achieving over $10\times$ faster inference. To optimally exploit the advantages of VAR for AiOR, we propose architectural modifications and improvements, including intricately designed cross-attention mechanisms and a latent-space refinement module, tailored for the AiOR task. Extensive experiments show that RestoreVAR achieves state-of-the-art performance among generative AiOR methods, while also exhibiting strong generalization capabilities. Project Page: `https://sudraj2002.github.io/restorevarpage/`

## 1 Introduction

Image restoration is a complex inverse problem that aims to recover clean images from degradations, such as haze, rain, snow, blur, and low-light conditions. Recently, the paradigm of All-in-One image Restoration (AiOR) has emerged, where a single network is trained to handle multiple degradation types. Existing AiOR methods can be broadly categorized into non-generative and generative approaches. Non-generative models such as AirNet (Li et al., 2022), PromptIR (Potlapalli et al., 2024), InstructIR (Conde et al., 2025), AWRaCLe (Rajagopalan & Patel, 2024), and AdaIR (Cui et al., 2024), deterministically map degraded images to their clean counterparts. While these methods offer fast inference and reliable pixel-level restoration performance, they often fail to generalize to diverse degradations encountered in real-world scenarios. To overcome this challenge, recent works have adopted generative models that aim to capture the distribution of clean images and produce more perceptually realistic outputs. Early works (Chen et al., 2022; Kupyn et al., 2018) based on GANs (Goodfellow et al., 2020) attempted this through adversarial learning, but suffered from mode collapse and unstable training. To improve fidelity and training stability, DiffUIR (Zheng et al., 2024) and DA-CLIP (Luo et al., 2023) employed pixel-space diffusion models (Ho et al., 2020). However, their high computational cost makes large-scale pretraining infeasible, limiting their ability to learn strong generative priors. In contrast, recent methods such as Diff-Plugin(Liu et al., 2024), AutoDIR(Jiang et al., 2023), and PixWizard (Lin et al., 2024) leverage latent diffusion models (LDMs), such as Stable Diffusion (Rombach et al., 2022). By operating in a latent space, LDMs significantly reduce computational costs, enabling large-scale pretraining which equips them with strong generative priors of natural images. These priors allow LDM-based AiOR methods to deliver perceptually realistic restoration and improved generalization to real-world degradations.

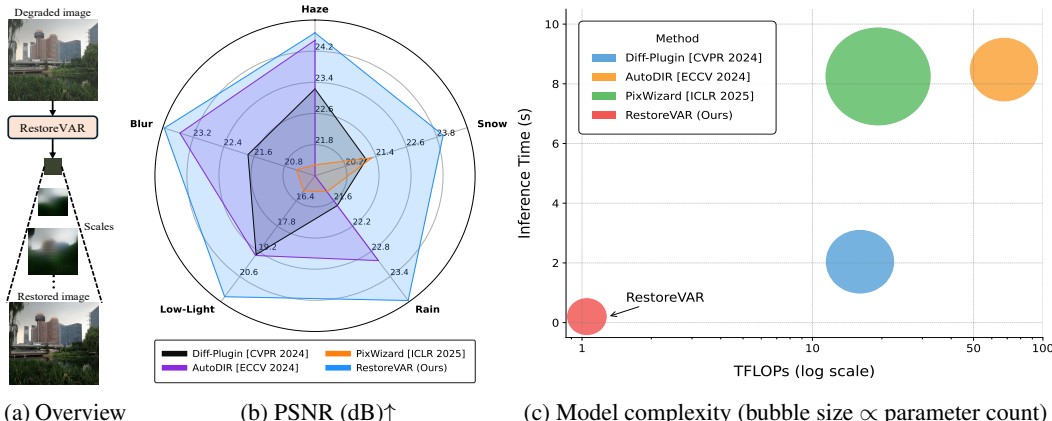

(a) Overview      (b) PSNR (dB)↑      (c) Model complexity (bubble size ∝ parameter count)

Figure 1: RestoreVAR, our proposed VAR-based (Tian et al., 2024) scale-space generative AiOR model (a), significantly outperforms LDM-based methods as shown in (b). RestoreVAR also offers drastic reductions in computational complexity as shown in (c).

Despite their advantages, LDM-based AiOR methods have some shortcomings. (1) LDMs require multiple denoising steps during inference, resulting in significantly longer runtimes compared to non-generative models. Their slow inference speeds pose challenges for applications that demand real-time processing, such as video surveillance or autonomous navigation. (2) LDMs rely on variational autoencoders (VAEs) (Kingma, 2013) which are primarily trained for generative diversity, rather than accurate pixel-level reconstruction. Consequently, the restored images obtained from LDM-based AiOR methods exhibit loss of fine structural details, hindering their performance.

Autoregressive models have driven rapid advances in natural language processing through large language models (LLMs) such as GPT-3 (Radford et al., 2019b) and LLaMA (Touvron et al., 2023). These models generate outputs by predicting the next token, conditioned on previously generated tokens. Recently, Visual AutoRegressive (VAR) Modeling (Tian et al., 2024) introduced scale-space autoregression for image generation, performing next-scale prediction in the latent space of a multi-scale vector-quantized VAE (VQVAE). VAR achieves performance comparable to state-of-the-art diffusion models such as DiT-XL/2 (Peebles & Xie, 2023), while operating $45\times$ faster. Despite its success in generative tasks, the application of VAR to low-level vision tasks such as image restoration remains largely unexplored. To the best of our knowledge, only two prior works: VarSR (Qu et al., 2025) and Varformer (Wang & Zhao, 2024), have used VAR for image restoration. VarSR focused exclusively on the super-resolution task, while Varformer utilized intermediate VAR features to guide a separate non-generative network for AiOR. In contrast, our approach is generative and fully exploits the strong priors of the pretrained VAR model by training it directly for the AiOR task. Our analysis in Sec. 3.2 also reveals that the scale-space decomposition of VAR captures degradations predominantly in coarse scales and scene-level details in fine scales, making it well-suited for AiOR.

To this end, we introduce RestoreVAR, a novel generative approach for AiOR that addresses some of the key limitations of LDM-based approaches. Firstly, RestoreVAR adopts the autoregressive structure of VAR, achieving state-of-the-art generative AiOR performance with over **10×** faster inference than LDM-based methods (see Fig. 1). Secondly, RestoreVAR employs cross-attention mechanisms conditioned on the degraded image latents, enabling the model to maintain spatial consistency and minimize hallucinations. Thirdly, to mitigate the loss of fine details by the vector quantization and VAE decoding processes, we propose a lightweight (only ∼ **3%** overhead) non-generative latent refinement transformer which predicts de-quantized latents from the outputs of VAR. Additionally, we fine-tune the VAE decoder to operate on these continuous latents, further enhancing reconstruction quality. Finally, through extensive experiments, we demonstrate that RestoreVAR achieves state-of-the-art performance among generative restoration models, while also exhibiting strong generalization to real-world degradations. To summarize, our key contributions are:

1. We propose RestoreVAR, the first VAR-based generative AiOR framework that achieves superior performance and a **10×** faster inference than LDM-based methods.
2. To achieve semantically coherent restoration, we introduce degraded image conditioning through cross-attention at each block of the VAR transformer.

3. To mitigate the loss of fine details in the vector quantization and VAE decoding processes, we introduce a non-generative latent refiner transformer which converts discretized latents into continuous ones, and fine-tune the VAE decoder to operate on continuous latents.

4. Extensive experiments show that RestoreVAR attains state-of-the-art performance among generative AiOR approaches, with perceptually preferable results and strong generalization.

## 2 RELATED WORKS

### 2.1 IMAGE RESTORATION

Early restoration models primarily addressed specific degradations (He et al., 2009; Zhang et al., 2020; Wang et al., 2019; Yasarla & Patel, 2019; Zhang et al., 2021a; Nah et al., 2017). Other methods such as Restormer (Zamir et al., 2022), MPRNet (Zamir et al., 2021) and SwinIR (Liang et al., 2021) introduced architectures for any single restoration task. However, they are restricted to handle one degradation at a time, making them ineffective for multiple degradations. All-in-One image Restoration (AiOR) methods aim to tackle multiple corruptions with a single model. Early approaches include non-generative models such as All-in-one (Li et al., 2020) and Transweather (Valanarasu et al., 2022). PromptIR (Potlapalli et al., 2024) used learnable prompts while AWRaCLe (Rajagopalan & Patel, 2024) utilized visual in-context learning to extract degradation characteristics. Other approaches such as InstructIR (Conde et al., 2025) adopted textual guidance, and DCPT (Hu et al., 2025) proposed a novel pre-training strategy for AiOR. DFPIR (Tian et al., 2025) proposed a feature perturbation strategy for AiOR. Recent AiOR methods have adopted diffusion models. Pixel-space diffusion models (PSDMs) such as DA-CLIP (Luo et al., 2023) and DiffUIR (Zheng et al., 2024) demonstrated improved AiOR performance but lacked robust generative priors. Recent methods have utilized the strong priors of LDMs for AiOR. Diff-Plugin (Liu et al., 2024) adopts task plugins to guide an LDM for AiOR. AutoDIR (Jiang et al., 2023) automatically detects and restores degradations using an LDM. PixWizard (Lin et al., 2024) is a multi-task Lumina-next (Zhuo et al., 2024) based model capable of performing AiOR among other tasks. However, LDM-based approaches are slow at inference time, a limitation we aim to overcome using visual autoregressive modeling (VAR).

### 2.2 AUTOREGRESSIVE MODELS IN VISION

Recent works (Van Den Oord et al., 2016; Tian et al., 2024) have extended autoregressive (AR) models to vision and can be categorized as pixel-space AR (Van Den Oord et al., 2016; Van den Oord et al., 2016; Chen et al., 2018b), token-based AR (Van Den Oord et al., 2017; Yu et al., 2023; Ramesh et al., 2021) and scale-space AR (Tian et al., 2024; Ren et al., 2024; Guo et al., 2025). Pixel-space AR predicts raw pixels one by one in raster order, as in PixelRNN (Van Den Oord et al., 2016) and PixelCNN++ (Salimans et al., 2017), but is very slow at high resolutions. Token-based AR compresses images into discrete latent codes via vector quantization (e.g., VQ-VAE (Van Den Oord et al., 2017), VQGAN (Esser et al., 2021)) and then models code sequences with transformers (e.g. ImageGPT (Chen et al., 2020)). This trades-off codebook size and transformer capacity against tractability for high-resolution generation. Scale-space AR, as introduced in VAR (Tian et al., 2024), generates latents from coarse to fine scales and matches the quality of Diffusion Transformers (Peebles & Xie, 2023) at a fraction of the inference cost. HART (Tang et al., 2024) scales VAR to higher resolution and uses a MLP-based diffusion refiner to convert discrete VAR latents into continuous representations. Despite VAR's success in generative tasks, it remains underexplored for image restoration with only two prior works-VarSR (Qu et al., 2025) and Varformer (Wang & Zhao, 2024). VarSR addressed super-resolution, while Varformer used VAR's features to guide a non-generative AiOR model. In contrast, RestoreVAR is a generative model which directly trains VAR for AiOR.

## 3 PROPOSED METHOD

We first explain the working principles behind VAR for image generation. We then describe our scale-space analysis of VAR and detail RestoreVAR, our proposed VAR-based approach for AiOR.

## 3.1 PRELIMINARIES: VISUAL AUTOREGRESSIVE MODELLING

Visual Autoregressive Modelling, or VAR, is a novel autoregressive class-conditioned image generation method which uses a GPT-2 (Radford et al., 2019a) style decoder-only transformer architecture for next-scale prediction. The VAR transformer operates in the latent space of a multi-scale VQVAE which uses $K$ scales.

Given an image $I \in \mathbb{R}^{H \times W \times 3}$, the VQ-VAE encoder outputs a latent representation $f_{\text{cont}} \in \mathbb{R}^{H_K \times W_K \times C}$. Hereafter, we will refer to $f_{\text{cont}}$ as the *continuous latent*, and the latent obtained after quantization as *discrete latent*. Instead of directly quantizing $f_{\text{cont}}$, a multi-scale residual quantization using a shared codebook across $K$ spatial scales is performed. First, the residual and accumulated quantized (or discrete) reconstruction of $f_{\text{cont}}$ are initialized as $f_{\text{res}}^{(0)} := f_{\text{cont}}$ and $f_{\text{quant}}^{(0)} := 0$, respectively. At each scale $k = 1, \ldots, K$, an index map $r_k \in \mathbb{Z}^{H_k \times W_k}$ is obtained by quantizing the downsampled residual feature:

$$r_k := \text{quantize}\left(\text{downsample}\left(f_{\text{res}}^{(k-1)}\right)\right).$$

The indices $r_k$ are then decoded using the codebook embeddings $e(\cdot)$, upsampled to match the full resolution, and refined using a convolutional module $\phi_k(\cdot)$ to obtain

$$h_k := \phi_k\left(\text{upsample}\left(e(r_k)\right)\right), \in \mathbb{R}^{H_K \times W_K \times C}.$$

This is done to approximate the information captured at the current scale which is used to update the residual continuous features to be modelled by subsequent scales as

Figure 2: VAR captures degradations in early scales (coarse) and scene-level details in later scales (fine). Degraded and GT are VQVAE reconstructions of the degraded and ground truth images. GT+coarse replaces early GT scales with degraded ones, while GT+fine replaces the late GT scales.

$$f_{\text{quant}}^{(k)} := f_{\text{quant}}^{(k-1)} + h_k, \quad f_{\text{res}}^{(k)} := f_{\text{cont}} - f_{\text{quant}}^{(k)}.$$

This process is repeated for all scales and yields a set of index maps $\{r_1, r_2, \ldots, r_K\}$, each consisting of the code-book indices of residual information at an increasingly finer scale.

For training, VAR uses teacher-forcing, where the ground-truth index maps $\{r_1, r_2, \ldots, r_K\}$ are used to autoregressively predict the next scale. For each scale $k$, the accumulated reconstruction $f_{\text{quant}}^{(k-1)} = \sum_{i=1}^{k-1} \phi_i\left(\text{upsample}(e(r_i))\right)$ is interpolated to the resolution of scale $k$ to obtain $\hat{f}_{\text{quant}}^{(k)}$, which is then flattened into tokens, and concatenated with the remaining tokens to form the input sequence. A start-of-sequence (SOS) token, derived from the class label embedding, is then prepended to this input sequence. A block-wise causal attention mask is used to ensure that predictions for scale $k$ attend only to the previous scales. VAR is trained to minimize the cross-entropy loss between predicted logits and the ground-truth index maps, modeling the likelihood

$$p(r_1, r_2, \ldots, r_K) = \prod_{k=1}^{K} p(r_k \mid r_1, r_2, \ldots, r_{k-1}).$$

During inference, the SOS token is created from the target class label. VAR then autoregressively predicts each index map $r_k$, one scale at a time. After predicting $r_k$, its embedding is upsampled, refined and accumulated to form the input for the next scale, mimicking the same procedure used during training. The VAR model uses only $K = 10$ latent scales with key-value (KV) caching, enabling significantly faster inference compared to latent diffusion models.

## 3.2 SCALE-SPACE ANALYSIS OF VAR

In addition to VAR's competitive performance to LDMs with far superior inference speed, we found that its residual scale-space decomposition focuses on degradations and scene-level details across different scales. To demonstrate this, we consider clean (GT)–degraded image pairs and compute

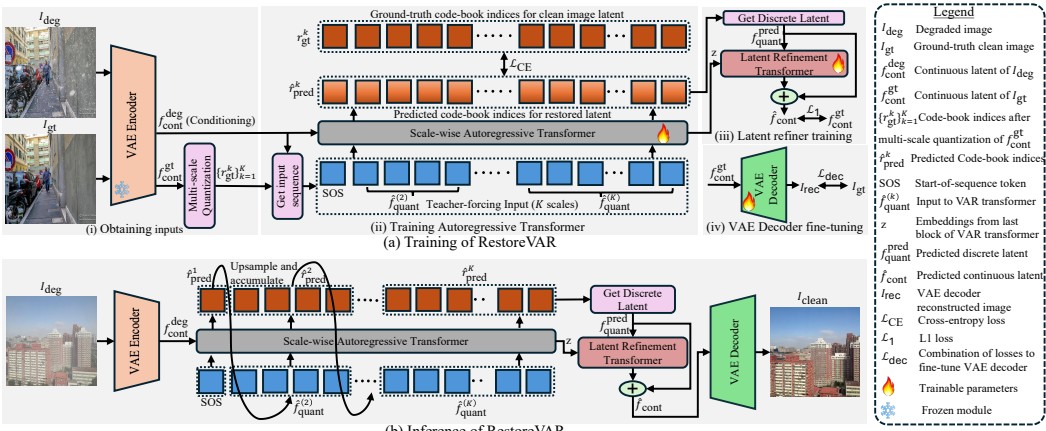

Figure 3: Illustration of RestoreVAR for training and inference. (a) Shows the training procedure for each component of RestoreVAR, and (b) shows the overall pipeline during inference.

their scale-wise residual indices $\{r_k^{\text{GT}}\}_{k=1}^K$ and $\{r_k^{\text{Deg}}\}_{k=1}^K$, respectively, where $K = 10$. We define coarse scales as $k = 1, \ldots, 5$ (low-resolution index maps) and fine scales as $k = 6, \ldots, 10$ (higher-resolution index maps). The first and last columns of Fig. 2 show reconstructions from $r_k^{\text{Deg}}$ and $r_k^{\text{GT}}$, respectively. In column 2, we replace the coarse scales of $r_k^{\text{GT}}$ with those from $r_k^{\text{Deg}}$. This introduces the degradation, although fine scales remain unchanged. Next, in column 3 we replace the fine scales of $r_k^{\text{GT}}$ with those from $r_k^{\text{Deg}}$, which yields a clean image with some loss of fine details. These observations indicate that coarse scales in VAR capture degradations, while finer scales encode scene-level detail. Notably, this observation holds across multiple degradations and simplifies restoration for VAR as removing degradations requires correctly predicting only the early scales which contain a small number of tokens, while scene details can be reconstructed in subsequent scales.

## 3.3 RESTOREVAR

We now describe RestoreVAR, our proposed approach that effectively adapts VAR for AiOR, leveraging its substantial inference speed advantage over LDMs. Given a degraded image $I_{\text{deg}} \in \mathbb{R}^{H \times W \times 3}$, the goal is to predict a clean output $I_{\text{clean}}$, close to the ground-truth $I_{\text{gt}}$. Adapting VAR to AiOR is non-trivial due to the need for high-quality pixel-level reconstruction, which is compromised by two factors: (1) VAR's strong generative priors can cause hallucinations in the restored images without proper conditioning. (2) Vector quantization and VAE decoding introduce artifacts that hinder pixel-level restoration. RestoreVAR addresses these challenges through architectural enhancements, including cross-attention to incorporate semantic guidance from the degraded image, and a novel non-generative transformer that refines discrete latents into their continuous form to preserve fine details in the restored image. We describe these components below.

### 3.3.1 AUTOREGRESSIVE TRANSFORMER ARCHITECTURE

For training, the multi-scale teacher-forcing input is constructed from the ground-truth image $I_{\text{gt}}$ (see Sec. 3.1). The start-of-sequence (SOS) token is computed from a fixed label index and augmented with a global context vector derived from the degraded image (see supplementary for details). These features are flattened and concatenated into a token sequence $\hat{f}_{\text{quant}} \in \mathbb{R}^{L \times C}$, where $L$ is the total number of tokens across all scales (see Fig. 3(a)(i)). The VAR transformer is then trained to autoregressively predict the next-scale indices $\{r_{\text{gt}}^k\}_{k=1}^K \in \mathbb{R}^L$ of the clean image.

To enable semantically consistent restoration, we inject information from the degraded image through cross-attention at each transformer block. At block $i$, the queries are given by the output of the feed-forward network ($x_{\text{block}_i} \in \mathbb{R}^{L \times D}$, where $D$ is the embedding dimension), while keys and values are derived from the continuous latent of the degraded image, $f_{\text{cont}}^{\text{deg}} \in \mathbb{R}^{H_K \times W_K \times C}$. This latent is reshaped into a sequence of conditioning tokens and is appropriately projected to the embedding dimension of the transformer. As shown in Sec. 4.4, conditioning on continuous latents significantly

outperforms conditioning on discrete ones. To summarize, cross-attention $(\mathrm{CA}(\cdot, \cdot))$ is applied as

$$x_{\mathrm{block_{CA}}} = x_{\mathrm{block}_i} + g_i \times \mathrm{CA}(x_{\mathrm{block}_i}, f_{\mathrm{cont}}^{\mathrm{deg}}).$$

We initialize $g_i = 0$ to retain VAR's pretrained behavior and gradually introduce conditioning. Furthermore, we replace absolute positional embeddings in VAR with 2D Rotary Positional Embeddings (RoPE) for scaling resolution from $256 \times 256$ to $512 \times 512$, as RoPE is well-suited for handling varying sequence lengths (Su et al., 2024). We also remove AdaLN layers, reducing $\sim 100\mathrm{M}$ parameters with negligible impact on performance. Inference closely follows that of VAR (see Sec. 3.1), except that each scale prediction is now guided by the degraded latent. The output is a sequence of predicted indices $\{\hat{r}_{\mathrm{pred}}^k\}_{k=1}^K$, which is then used to construct the discrete restored latent $f_{\mathrm{quant}}^{\mathrm{pred}} \in \mathbb{R}^{H_K \times W_K \times C}$. The above steps are shown in Fig. 3(a)(ii). More architectural details are given in the supplementary.

### 3.3.2 DETAIL-PRESERVING RESTORATION

The discrete latent ($f_{\mathrm{quant}}^{\mathrm{pred}}$) predicted by the RestoreVAR transformer is decoded by the VQVAE to produce the restored image. However, vector-quantization and VAE decoding cause a noticeable loss of fine details in the pixel-space, leading to distorted reconstructions. This presents a major challenge for using VAR in AiOR, as the scene semantics may not be accurately preserved. To address this, we introduce VAE decoder fine-tuning on continuous latents, and a lightweight latent refinement transformer (LRT) that converts discrete latents to continuous latents for decoding.

**VAE Decoder Fine-Tuning.** HART (Tang et al., 2024) addressed VAE-induced distortions by fine-tuning the VAE decoder on both discrete and continuous latents. While effective for generative tasks, the VAE decoder of HART produces overly textured outputs, compromising accurate reconstruction (see supplementary). Instead, we fine-tune the decoder only on continuous latents, bypassing the quantizer. The encoder and quantizer are kept frozen, and the decoder is trained on $(f_{\mathrm{cont}}^{\mathrm{gt}}, I_{\mathrm{gt}})$ pairs. To avoid overly smooth outputs, we use a PatchGAN (Isola et al., 2017) discriminator (see Sec. 4.4) and optimize the decoder using pixel-wise, perceptual, and adversarial losses as

$$\mathcal{L}_{\mathrm{dec}} = \lambda_1 \mathcal{L}_{\mathrm{L1}} + \lambda_2 \mathcal{L}_{\mathrm{SSIM}} + \lambda_3 \mathcal{L}_{\mathrm{percep}} + \lambda_4 \mathcal{L}_{\mathrm{adv}},$$

where $\mathcal{L}_{\mathrm{L1}}$ is the L1 loss, $\mathcal{L}_{\mathrm{SSIM}}$ is the SSIM loss, $\mathcal{L}_{\mathrm{percep}}$ is the perceptual loss, $\mathcal{L}_{\mathrm{adv}}$ is the adversarial loss and $\lambda_i$ are their respective weights (see Fig. 3(a)(iv)). Our fine-tuning approach yields a decoder that is well-aligned with the objectives of AiOR, achieving mean (over 1000 samples) reconstruction PSNR/SSIM scores of 28.14dB/0.842, outperforming both the VAR VQVAE (22.59dB/0.679) and HART decoders (26.48dB/0.804). Qualitative comparisons are given in the supplementary.

**Refining Discrete Latents.** Since the VAE decoder is fine-tuned for continuous latents, the predicted discrete latent, $f_{\mathrm{quant}}^{\mathrm{pred}}$, must be converted into a continuous form for decoding. While HART uses a 37M parameter diffusion-based MLP for this, it incurs a $\sim 20\%$ inference overhead due to iterative denoising. Instead, we propose a lightweight, non-generative latent refinement transformer (LRT) that predicts a residual, which when added to $f_{\mathrm{quant}}^{\mathrm{pred}}$, produces a continuous latent, $\hat{f}_{\mathrm{cont}} \in \mathbb{R}^{H_K \times W_K \times C}$ as

$$\hat{f}_{\mathrm{cont}} = f_{\mathrm{quant}}^{\mathrm{pred}} + \mathrm{LRM}(f_{\mathrm{quant}}^{\mathrm{pred}}, z),$$

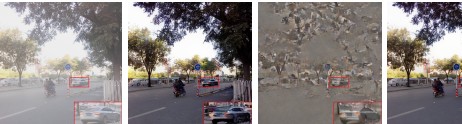

| Input | Discrete | Refiner | Continuous |

Figure 4: Illustration of images decoded from discrete and continuous latents, along with the refiner's predicted residuals.

where $z \in \mathbb{R}^{L \times D}$ is the output from the final RestoreVAR transformer block. $z$ is passed through cross-attention and provides pseudo-continuous guidance to the LRT which is critical for performance (see Sec. 4.4). The LRT is trained using $\mathcal{L}_1$ loss between the predicted and ground-truth continuous latents ($f_{\mathrm{cont}}^{\mathrm{gt}}$) as $\mathcal{L}_{\mathrm{LRT}} = \mathcal{L}_1(\hat{f}_{\mathrm{cont}}, f_{\mathrm{cont}}^{\mathrm{gt}})$. Our LRT introduces only 3% additional overhead and significantly outperforms HART's refiner in PSNR and SSIM scores (see Sec. 4.4). The training procedure of the LRT is shown in Fig. 3(a)(iii) and Fig. 4 provides a visual example of its predictions.

Thus, RestoreVAR combines the VAR transformer, LRT, and fine-tuned decoder to deliver fast, perceptually realistic, and structurally faithful results. Fig. 3(b) depicts inference of RestoreVAR.

## 4 EXPERIMENTS

In this section, we provide implementation details, comparisons with existing All-in-One image Restoration (AiOR) approaches, and present ablations on key components of our framework.

Table 1: Quantitative comparisons of RestoreVAR with the state-of-the-art LDM-based generative AiOR approaches, and non-generative methods. RestoreVAR significantly outperforms generative methods on PSNR, SSIM and LPIPS scores. The best generative approach is indicated in **bold**.

| Method | Venue | RESIDE | | | Snow100k | | | Rain13K | | | LOLv1 | | | GoPro | | |
|---|---|---|---|---|---|---|---|---|---|---|---|---|---|---|---|---|
| | | PSNR↑ | SSIM↑ | LPIPS↓ | PSNR↑ | SSIM↑ | LPIPS↓ | PSNR↑ | SSIM↑ | LPIPS↓ | PSNR↑ | SSIM↑ | LPIPS↓ | PSNR↑ | SSIM↑ | LPIPS↓ |
| *Non-generative methods* | | | | | | | | | | | | | | | | |
| PromptIR | NeurIPS'23 | 32.02 | 0.952 | 0.013 | 31.98 | 0.924 | 0.115 | 29.56 | 0.888 | 0.087 | 22.89 | 0.847 | 0.296 | 27.21 | 0.817 | 0.250 |
| InstructIR | ECCV'24 | 26.90 | 0.952 | 0.017 | – | – | – | 29.56 | 0.885 | 0.088 | 22.81 | 0.836 | 0.132 | 28.26 | 0.870 | 0.146 |
| AWRaCLe | AAAI'25 | 30.81 | 0.979 | 0.013 | 30.56 | 0.904 | 0.088 | 31.26 | 0.908 | 0.068 | 21.04 | 0.818 | 0.146 | 26.78 | 0.820 | 0.248 |
| DCPT | ICLR'25 | 29.10 | 0.968 | 0.017 | – | – | – | 24.11 | 0.766 | 0.203 | 23.67 | 0.863 | 0.106 | 27.92 | 0.877 | 0.169 |
| DFPIR | CVPR'25 | 31.39 | 0.979 | 0.012 | – | – | – | 24.87 | 0.794 | 0.171 | 23.12 | 0.853 | 0.123 | 28.66 | 0.884 | 0.158 |
| *Generative methods* | | | | | | | | | | | | | | | | |
| Diff-Plugin | CVPR'24 | 23.23 | 0.765 | 0.091 | 21.02 | 0.611 | 0.196 | 21.71 | 0.617 | 0.169 | 19.38 | 0.713 | 0.195 | 21.76 | 0.633 | 0.217 |
| AutoDIR | ECCV'24 | 24.48 | 0.780 | 0.081 | 19.00 | 0.515 | 0.347 | 23.02 | 0.642 | 0.162 | 19.43 | 0.766 | 0.135 | 23.55 | 0.700 | 0.168 |
| PixWizard | ICLR'25 | 21.28 | 0.738 | 0.142 | 21.24 | 0.594 | 0.206 | 21.38 | 0.596 | 0.180 | 15.84 | 0.629 | 0.305 | 20.49 | 0.602 | 0.223 |
| **RestoreVAR (Ours)** | | **24.67** | **0.821** | **0.074** | **24.05** | **0.713** | **0.156** | **23.97** | **0.700** | **0.153** | **21.72** | **0.782** | **0.126** | **23.96** | **0.737** | **0.167** |

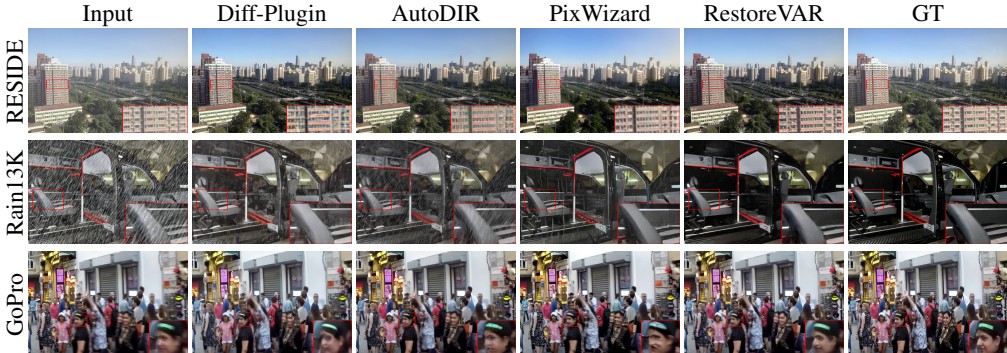

Figure 5: Qualitative comparisons of RestoreVAR with LDM-based generative AiOR approaches. RestoreVAR achieves consistent restoration with enhanced preservation of fine-details.

## 4.1 IMPLEMENTATION DETAILS

Each component of RestoreVAR was trained independently to disentangle learning objectives. We used the VAR model of depth 16 as the transformer backbone and trained it with the AdamW optimizer (Loshchilov & Hutter, 2017), a learning rate (LR) of $10^{-4}$, batch size of 48, for 100 epochs. The latent refiner was trained for 100 epochs with the AdamW optimizer, LR= $10^{-4}$ and a batch size of 96. The VAE decoder was fine-tuned using a weighted loss combination (see Sec. 3.3.2) with empirically chosen weights: $\lambda_1 = 2.0$, $\lambda_2 = 0.4$, $\lambda_3 = 0.2$, and $\lambda_4 = 0.01$. Fine-tuning was performed for 5 epochs with a learning rate of $3 \times 10^{-4}$ and a batch size of 12, using AdamW. Training was conducted on 8 RTX A6000 GPUs, while inference was done on an RTX 4090 GPU.

## 4.2 DATASETS

We trained RestoreVAR for five tasks: dehazing, desnowing, deraining, low-light enhancement and deblurring. For dehazing, we used the RESIDE (Li et al., 2019) dataset comprising 72135 training and 500 test images. The Snow100k dataset (Liu et al., 2018) was used for desnowing, with 50000 training and 16801 test images (heavy subset). For deraining, we used Rain13K (Zamir et al., 2021) consisting of 13711 training and 4298 test images. The LOLv1 (Wei et al., 2018) dataset was used for low-light enhancement, consisting of 485 training and 15 test images. For deblurring, we used the GoPro (Nah et al., 2017) dataset comprising 2103 training and 1111 test images. We also assess generalization performance on real-world, unseen and mixed degradation datasets, namely, LHP (Guo et al., 2023) (1000 images), REVIDE (Zhang et al., 2021b) (284 images), TOLED (Zhou et al., 2021) (30 images), POLED (Zhou et al., 2021) (30 images), CDD (Guo et al., 2024) (200 images, mix of haze and rain), and LOLBlur (Zhou et al., 2022) (482 images, mix of low-light and blur). TOLED and POLED datasets contain unseen degradation of under-display camera restoration.

## 4.3 COMPARISONS

We compare RestoreVAR with state-of-the-art generative and non-generative methods for AiOR. For non-generative approaches, we include PromptIR (Potlapalli et al., 2024), InstructIR (Conde et al.,

Table 2: Quantitative comparisons of RestoreVAR against state-of-the-art non-generative and generative approaches on real-world, unseen and mixed degradations. The best result among RestoreVAR and non-generative approaches is indicated in bold.

| Method | LHP | | REVIDE | | TOLED | | POLED | | LOLBlur (L + B) | | CDD (H + R) | | Average | |
|---|---|---|---|---|---|---|---|---|---|---|---|---|---|---|
| | MUSIQ↑ | CLIPIQA↑ | MUSIQ↑ | CLIPIQA↑ | MUSIQ↑ | CLIPIQA↑ | MUSIQ↑ | CLIPIQA↑ | MUSIQ↑ | CLIPIQA↑ | MUSIQ↑ | CLIPIQA↑ | MUSIQ↑ | CLIPIQA↑ |
| PromptIR | 56.780 | 0.366 | 61.191 | 0.459 | 43.218 | 0.281 | 34.536 | 0.303 | 33.693 | 0.166 | 65.895 | 0.483 | 49.219 | 0.343 |
| InstructIR | **58.269** | 0.359 | 63.116 | 0.416 | 44.985 | 0.298 | 23.317 | 0.241 | 40.221 | 0.202 | 65.491 | 0.482 | 49.900 | 0.333 |
| AWRaCLe | 57.889 | 0.333 | 59.287 | 0.368 | 44.670 | 0.285 | 40.533 | 0.332 | 38.186 | 0.171 | 66.253 | 0.484 | 51.470 | 0.329 |
| DCPT | 58.044 | 0.372 | 60.011 | 0.446 | 44.062 | 0.314 | 38.138 | **0.345** | 37.393 | 0.175 | 68.440 | 0.544 | 51.681 | 0.366 |
| DFPIR | 56.483 | 0.330 | 61.009 | 0.450 | 43.820 | 0.276 | 35.668 | 0.289 | 36.277 | 0.163 | 54.408 | 0.349 | 47.611 | 0.310 |
| DiffPlugin | 57.351 | 0.420 | 63.483 | 0.406 | 46.219 | 0.311 | 35.086 | 0.407 | 40.054 | 0.212 | 68.791 | 0.578 | 51.831 | 0.389 |
| AutoDir | 58.085 | 0.380 | 63.918 | 0.416 | 54.585 | 0.341 | 48.796 | 0.380 | 46.642 | 0.225 | 68.575 | 0.571 | 56.767 | 0.386 |
| PixWizard | 58.600 | 0.411 | 68.487 | 0.420 | 45.804 | 0.329 | 44.305 | 0.348 | 50.708 | 0.268 | 68.409 | 0.589 | 56.052 | 0.394 |
| **RestoreVAR** | 57.662 | **0.414** | **63.562** | **0.483** | **52.374** | **0.338** | **48.118** | 0.276 | **46.644** | **0.214** | **68.941** | **0.572** | **56.217** | **0.383** |

2025), AWRaCLe (Rajagopalan & Patel, 2024), DCPT (Hu et al., 2025) and DFPIR (Tian et al., 2025). Among generative methods, we compare with the LDM-based approaches Diff-Plugin (Liu et al., 2024), AutoDIR (Jiang et al., 2023) and PixWizard (Lin et al., 2024). To ensure a fair comparison, we retrained PromptIR and AWRaCLe, as their official checkpoints were not trained for most of our AiOR tasks. All other methods were evaluated using their publicly released checkpoints. For AutoDIR, we report results without the structure correction module, as this module functions as an independent, non-generative restoration network (more details in supplementary). The results reported for PixWizard were obtained using its publicly released checkpoint. We do not compare with task-specific restoration models, as RestoreVAR is proposed for the AiOR setting.

Table 1 presents PSNR, SSIM and LPIPS scores on the RESIDE, Snow100k, Rain13K, LOLv1 and GoPro datasets. RestoreVAR surpasses LDM-based AiOR methods at a fraction of their computational cost (inference time (s) per image)—Diff-Plugin:2.04s, AutoDIR: 8.477s, PixWizard: 8.247s and RestoreVAR: 0.201s, highlighting the efficacy of our framework. More detailed complexity comparisons are given in the supplementary along with a derivation showing that the time complexity of VAR with maximum latent resolution $n \times n$ is $\mathcal{O}(\log n)$ lower than an LDM operating at the same latent resolution. Qualitative comparisons with LDM-based methods in Fig. 5 further illustrate that RestoreVAR produces restored images of high quality while better preserving fine details. Visual results for the Snow100k and LOLv1 datasets are provided in the supplementary. While non-generative methods achieve better scores, it is important to recognize that the performance of RestoreVAR is inherently influenced by the quality of the VAE decoder; a limitation shared by all latent generative approaches. Despite this constraint, RestoreVAR narrows the gap with non-generative methods while maintaining the benefits of a generative framework, i.e., perceptually realistic results and strong generalization capabilities. To demonstrate these strengths, we evaluate generalization using no-reference image quality metrics (following prior works (Liu et al., 2024; Jiang et al., 2023; Rajagopalan & Patel, 2024)), and assess perceptual realism through a user study.

For testing generalization, we report MUSIQ (Ke et al., 2021) and CLIP-IQA (Wang et al., 2023) scores in Table 2 on the real-world, unseen and mixed degradation datasets discussed in Sec. 4.2. RestoreVAR achieves higher scores than non-generative models (on average), indicating better robustness under these degradations. Qualitative results for this experiment are shown in Fig. 6, where RestoreVAR consistently outperforms non-generative approaches. Due to space constraints, qualitative comparisons with PromptIR and visual results for LOLBlur are given in the supplementary. Furthermore, RestoreVAR achieves performance competitive to those of LDM-based meth-

Table 3: Mean scores from user study.

| Method | Score ↑ |
|---|---|
| PromptIR | 2.11 |
| InstructIR | 2.93 |
| AWRaCLe | 2.33 |
| DCPT | 2.42 |
| DFPIR | 2.35 |
| AutoDIR | 3.68 |
| **RestoreVAR** | **4.36** |

ods on these datasets. The slightly lower metrics for RestoreVAR can be explained by the fact that the latent diffusion backbones (Stable Diffusion (Rombach et al., 2022)) used by the competing generative models are trained on substantially larger datasets, typically hundreds of millions of images, whereas the VAR backbone is trained on only $\sim$ 1M ImageNet-1K (Deng et al., 2009) images. Despite this, RestoreVAR demonstrates comparable generalization performance while offering much faster inference and improved pixel-level fidelity. To further evaluate perceptual quality, we conducted a user study in which participants rated outputs from non-generative models, AutoDIR (LDM-based) and RestoreVAR, for 50 real-world scenes. We received 36 responses with each participant scoring outputs based on scene consistency, restoration quality, and overall appeal on a 5-point scale. Table 3 shows that RestoreVAR received the highest average ratings (across all three criteria), highlighting its ability to produce images that align closely with human preferences. More quantitative results are provided in the supplementary.

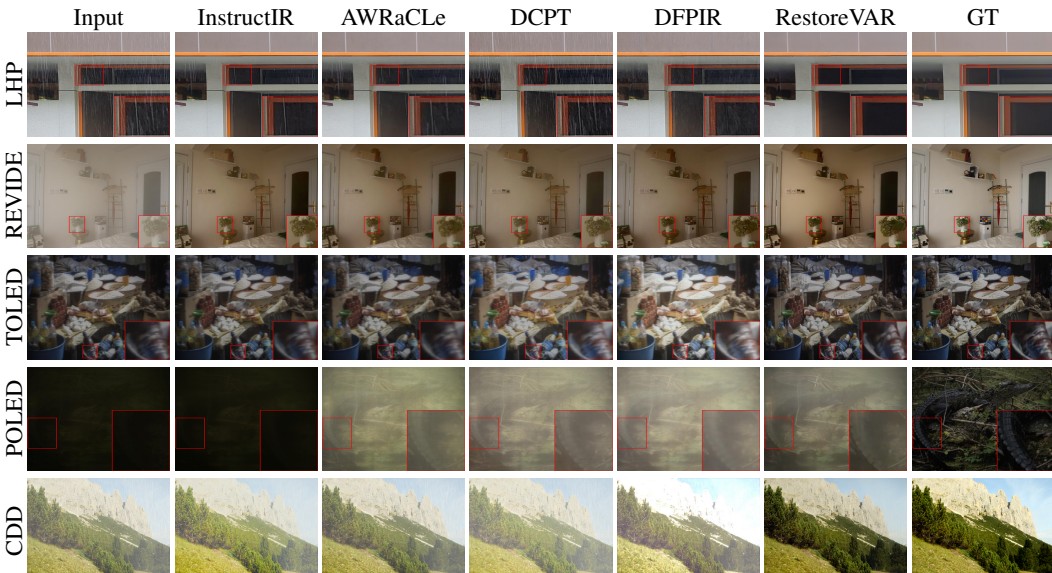

Figure 6: Qualitative comparisons of RestoreVAR with non-generative methods on real, unseen and mixed degradations. RestoreVAR consistently achieves better results.

To summarize, although non-generative AiOR models offer fast inference and good pixel-level fidelity, their generalization to real-world, unseen, and mixed degradations is limited. They work well only when test samples come from a distribution close to their training data, which hinders real-world applicability. In contrast, LDM-based generative approaches exhibit substantially stronger perceptual quality and generalization, but they are computationally expensive and suffer from low pixel-level fidelity. RestoreVAR addresses the limitations of LDM-based approaches while retaining their strengths. It achieves far superior pixel-level fidelity as established by the multi-task restoration performance on datasets in Table 1. Furthermore, on the real-world and unseen degradation datasets (Table 2), RestoreVAR achieves perceptual quality on par with diffusion-based methods while significantly outperforming non-generative models, demonstrating strong generalization. Moreover, RestoreVAR is substantially faster than LDM-based approaches and even nears the efficiency of some non-generative methods (see supplementary for details), despite requiring multiple steps for inference. Together, these results demonstrate that RestoreVAR closes the fidelity gap between LDM-based generative models and non-generative approaches while retaining strong perceptual quality and generalization, and offering enhanced computational efficiency.

We discuss limitations of RestoreVAR in the supplementary.

## 4.4 ABLATIONS AND ANALYSIS

**Continuous vs. Discrete Conditioning.** The RestoreVAR transformer conditions on the continuous latent of the degraded image ($f_{cont}^{deg}$). While conditioning with discrete multi-scale latents appears more aligned with VAR's multi-scale prediction objective, it results in significantly worse performance. To demonstrate this, we train RestoreVAR with discrete and continuous conditioning for 15 epochs each. As shown in Fig. 7, RestoreVAR with discrete conditioning exhibits much lower validation accuracy.

Table 4: Ablations on the types of latent refiners. Our proposed latent refiner transformer (LRT) performs best, with minimal overhead.

| Refiner Type | Time (s) | Params (M) | PSNR / SSIM |
|---|---|---|---|
| No Refiner | – | – | 21.71 / 0.690 |
| HART Refiner | 0.0455 | 36.06 | 23.48 / 0.777 |
| LRT w/o Last-Block | 0.0036 | 14.61 | 21.23 / 0.660 |
| Proposed LRT | 0.0061 | 22.97 | 24.67 / 0.821 |

**Discriminator for VAE fine-tuning.** As described in Sec. 3.3.2, we fine-tune the VAE decoder on continuous latents using a combination of pixel-level loss and an adversarial loss. To analyze the impact of the discriminator, we compare the reconstructions of VAE decoders fine-tuned with and without the adversarial loss. As shown in Fig. 8, removing the discriminator leads to blurrier reconstruction while including it yields sharper and perceptually better looking outputs. Ablations on SSIM and perceptual losses are provided in the supplementary.

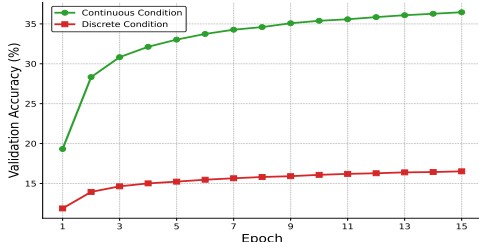

Figure 7: Validation accuracy of RestoreVAR under discrete vs. continuous conditioning.

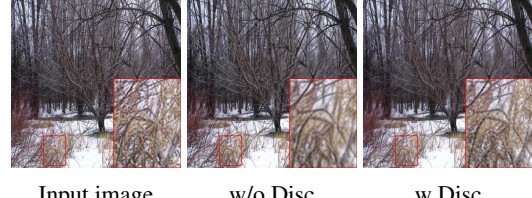

Input image     w/o Disc     w Disc

Figure 8: Image reconstructed by VAE decoders fine-tuned on continuous latents with (w) and without (w/o) a discriminator (Disc).

**Latent Refiner Transformer.** The Latent Refiner Transformer (LRT) is critical for preserving pixel-level detail in restored images. To analyze its impact, we compare four RestoreVAR variants: (i) No refiner, (ii) HART's diffusion refiner, (iii) LRT without final block outputs, and (iv) our proposed LRT. As shown in Table 4, our LRT achieves the best PSNR and SSIM, while maintaining low inference time and a low parameter count. Using no refiner yields poor PSNR/SSIM scores.

Removing the last block outputs significantly reduces performance, indicating its importance as pseudo-continuous guidance for refinement. HART's MLP diffusion-based refiner performs worse than our LRT while having a much higher parameter count and runs $\sim 7\times$ slower. To assess if the LRT causes loss of perceptual quality, we calculated MUSIQ and CLIPIQA scores (higher is better) with and without the LRT. Using no LRT yields a MUSIQ/CLIPIQA score of $66.56/0.472$ while incorporating the LRT results in a score of $66.45/0.481$. The scores are nearly identical indicating negligible impact on perceptual quality.

**Scale-space analysis of outputs.** A core motivation behind our approach is that the scale-space residual quantization of VAR captures degradations at coarse scales and scene level details at finer scales (see Sec. 3.2). Fig. 9 visualizes this for the outputs of RestoreVAR by comparing (i) the degraded image, (ii) the restored image with its coarse scales replaced by those of the degraded input, (iii) the re-

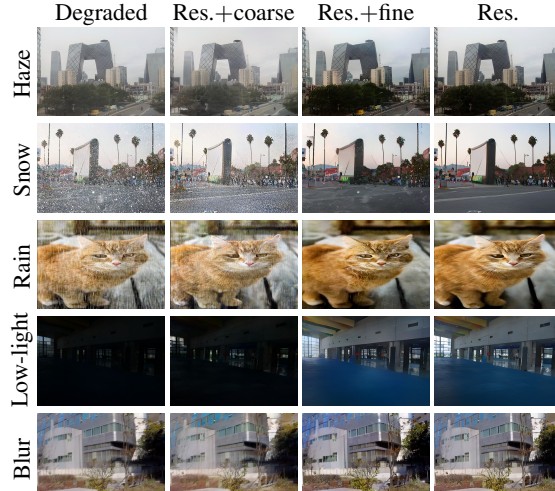

Degraded   Res.+coarse   Res.+fine    Res.

Figure 9: Scale-space analysis of outputs from RestoreVAR. Res. denotes the Restored output. Res.+coarse replaces early scales of restored output with degraded ones, while Res.+fine replaces the late scales. It can be observed that the coarse scales contribute most to overall restoration.

stored image with only fine scales replaced, and (iv) the restored image. Similar to our observations in Sec. 3.2, replacing the coarse-scale components of the restored output reintroduces the degradation, whereas replacing only the finer scales does not. This once again demonstrates that degradations predominantly reside in the coarse-scale indices, and that RestoreVAR predicts coarse-scale tokens that are mostly free of degradation while reconstructing scene-level details at the finer scales.

More ablations are provided in the supplementary.

## 5 CONCLUSIONS

We proposed RestoreVAR, a fast and effective generative approach for AiOR. Built on the VAR backbone, RestoreVAR benefits from VAR's strong generative priors and significantly faster inference compared to LDMs. To tailor VAR for AiOR, we introduced cross-attention mechanisms that inject semantic information from the degraded image into the generation process. Additionally, we proposed a non-generative latent refiner transformer to convert discrete latents to continuous ones, along with a VAE decoder fine-tuned on continuous latents, which together improve reconstruction fidelity. RestoreVAR achieves state-of-the-art performance among generative AiOR models, outperforming LDM-based methods while delivering over $10\times$ faster inference and strong generalization.

ACKNOWLEDGMENTS

This work is supported by the Intelligence Advanced Research Projects Activity (IARPA) via Department of Interior/ Interior Business Center (DOI/IBC) contract number 140D0423C0076. The U.S. Government is authorized to reproduce and distribute reprints for Governmental purposes notwithstanding any copyright annotation thereon. Disclaimer: The views and conclusions contained herein are those of the authors and should not be interpreted as necessarily representing the official policies or endorsements, either expressed or implied, of IARPA, DOI/IBC, or the U.S. Government.

ETHICS STATEMENT

We acknowledge that we have read and adhered to the ICLR Code of Ethics. For all our experiments, we used publicly available open-source datasets.

REPRODUCIBILITY STATEMENT

Our code will be made publicly available after the review process. Details to reproduce the work have been provided in Secs. 3 and 4.1 of the main paper and Sec. F of the supplementary.

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

## A    OVERVIEW OF SUPPLEMENTARY

In this supplementary, we first present detailed computational complexity comparisons of RestoreVAR and LDM-based AiOR approaches. We then provide a theoretical analysis comparing the time complexities of VAR and LDMs. Subsequently, we analyze the effect of using Absolute Positional Embeddings (APE) versus Rotary Positional Embeddings (RoPE) (Su et al., 2024) when scaling the resolution from $256 \times 256$ to $512 \times 512$. Next, we present more architectural details of RestoreVAR, followed by a breakdown of the runtime and parameter count for each component of RestoreVAR. We then provide additional visual results, which include qualitative results for the continuous vs. discrete conditioning ablation, visual comparisons of VAE decoders, refiner ablation, and more qualitative comparisons with other methods. Subsequently, we provide experiments to show that the structure correction module of AutoDIR (Jiang et al., 2023) behaves as an independent non-generative restoration network. We then discuss the limitations of our approach and scope for future work. Finally, we mention the usage of LLMs in the paper. To summarize, the supplementary discusses the following:

1. Detailed computation complexity comparisons (Sec. B)

2. Theoretical complexity comparison with LDM (Sec. C)

3. Additional Ablations (Sec. D)

    (a) Performance analysis: APE vs. RoPE (Sec. D.1)

    (b) Additional VAE loss ablations (Sec. D.2)

    (c) Additional scale-space analysis (Sec. D.3)

    (d) Additional LRT analysis (Sec. D.4)

4. Additional Quantitative Results (Sec. E)

    (a) User Study (Sec. E.1)

    (b) Generalization Metrics (Sec. E.2)

    (c) More Perceptual Quality Evaluations (Sec. E.3)

5. Additional Architectural Details (Sec. F)

6. Runtime and Parameter Breakdown (Sec. G)

7. Additional Visual Results (Sec. H)

    (a) Continuous vs. Discrete Conditioning (Sec. H.1)

    (b) Qualitative comparisons of VAE decoders (Sec. H.2)

    (c) Visual Comparison of Refiner Variants (Sec. H.3)

    (d) Additional Qualitative Comparisons (Sec. H.4)

8. More details about AutoDIR comparison (Sec. I)

9. Limitations and scope for future work (Sec. J)

10. LLM Usage (Sec. K)

## B    DETAILED COMPUTATION COMPLEXITY COMPARISONS

RestoreVAR achieves substantial performance improvements over LDM-based AiOR approaches at a fraction of their computational cost. To show this, we compared RestoreVAR with Diff-Plugin (Liu et al., 2024), AutoDIR (Jiang et al., 2023), and PixWizard (Lin et al., 2024) in terms of inference steps, runtime, TeraFLOPs, and total parameter count. As shown in Table 5, RestoreVAR achieves a $\mathbf{10\times}$ speed-up over Diff-Plugin and a $\sim \mathbf{16\times}$ reduction in TFLOPs. Compared to AutoDIR and PixWizard, RestoreVAR is over $\mathbf{40\times}$ faster in inference.

Table 5: Comparison of the computational complexity of RestoreVAR with LDM-based AiOR approaches.

| Method | Steps | Time (s) | TFLOPs | Params (M) |
|---|---|---|---|---|
| Diff-Plugin | 20 | 2.04 | 16.08 | 859.50 |
| AutoDIR | 100 | 8.477 | 67.80 | 859.50 |
| PixWizard | 60 | 8.247 | 19.27 | 2011.40 |
| RestoreVAR | 10 | 0.201 | 1.05 | 296.95 |

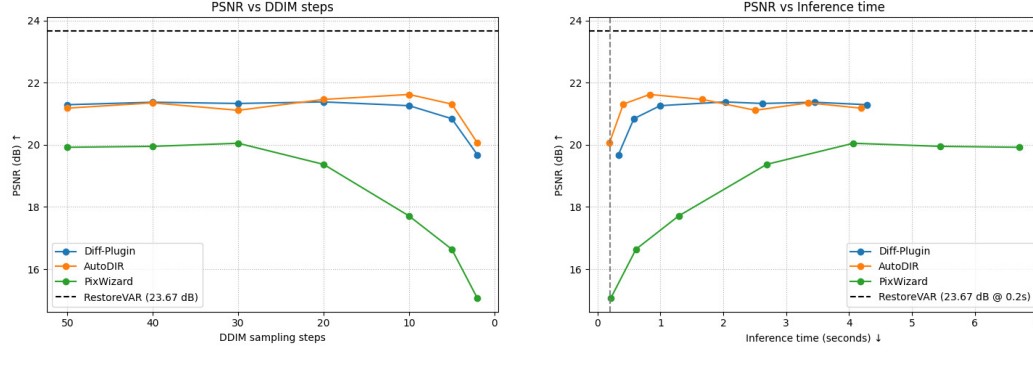

(a) PSNR vs DDIM steps  (b) PSNR vs inference time

Figure 10: Comparison of LDM-based methods accelerated using DDIM sampling and RestoreVAR. RestoreVAR achieves best results.

Additionally, we conducted an experiment to speed up LDM sampling using DDIM Song et al. (2020) sampling and compared the performance with RestoreVAR. Specifically, we varied the number of DDIM sampling steps as $50, 40, 30, 20, 10, 5$ and $2$. Figs. 10(a) and (b) shows the variation of mean (across the datasets from Table 1) PSNR (dB) scores with inference time (seconds) and DDIM sampling steps. Diffusion models match the inference time of RestoreVAR only when using around 2 sampling steps, but show over a 3dB decrease in PSNR compared to RestoreVAR. Even at higher step counts with DDIM sampling, diffusion models lag by over 1dB, highlighting RestoreVAR's clear advantages in both speed and performance over LDM-based methods.

Consistency models and rectified-flow approaches are promising recent developments in accelerating diffusion models. However, both require substantial modifications to the training pipeline. Consistency models demand fine-tuning pre-trained diffusion models with a new consistency objective that enables few-step inference. However, using very few steps can cause over-smoothed outputs that are undesirable for image restoration (Wang et al., 2025). Rectified-flow methods learn a deterministic velocity field along a straight-line path between noise and data, using a flow-matching objective. Although these frameworks speed up inference, they still lack the scale-space (Sec. 3.2) insight offered by VAR, which elegantly fits the objective of all-in-one image restoration.

We also provide computational complexity comparisons with non-generative methods in Table 6. Non-generative models are generally faster since RestoreVAR, like other generative approaches, requires multiple inference steps. Nevertheless, RestoreVAR maintains competitive inference speed to methods such as PromptIR, AWRaCLe, and DFPIR, even though it contains more parameters and requires a multi-step generation process. While it is slower than methods such as DCPT and InstructIR, the substantially improved generalization and superior perceptual quality offered by RestoreVAR make it a more robust choice.

Table 6: Comparison of the computational complexity of RestoreVAR with non-generative approaches.

| Method | Time (s) | Params (M) |
|---|---|---|
| PromptIR | 0.162 | 35.59 |
| InstructIR | 0.023 | 15.84 |
| AWRaCLe | 0.188 | 186.68 |
| DCPT | 0.024 | 67.88 |
| DFPIR | 0.117 | 182.38 |
| RestoreVAR | 0.201 | 296.95 |

## C  THEORETICAL COMPLEXITY COMPARISON WITH LDM

We now provide a theoretical comparison of the run-time complexities of VAR and diffusion transformers (DiT), offering fundamental insights into their efficiency differences. The VAR time complexity derivation closely follows that in the original VAR paper.

Let $a > 1$ be the geometric factor of the vector quantized (VQ) scale pyramid and let the largest scale have dimensions $h = w = n$. Let the number of scales be $K = \log_a n + 1$ so that the side length

at scale $i$ is $n_i = a^{i-1}$ and the largest scale is $n_K = n = a^{K-1}$. Assume a standard self-attention transformer as in VAR with time complexity $\mathcal{O}(T^2)$ for $T$ tokens.

**VAR.** At the generation of the $k$-th scale the total number of tokens across the current and previous scales $(r_1, \ldots, r_k)$ is

$$\sum_{i=1}^{k} n_i^2 = \sum_{i=1}^{k} a^{2(i-1)} = \frac{a^{2k} - 1}{a^2 - 1}.$$

Hence the cost of generation of the $k$-th scale is

$$C_k^{\text{VAR}} = \left( \frac{a^{2k} - 1}{a^2 - 1} \right)^2.$$

Summing over all $K$ scales gives

$$C^{\text{VAR}} = \sum_{k=1}^{K} \left( \frac{a^{2k} - 1}{(a^2 - 1)} \right)^2 = \frac{1}{(a^2 - 1)^2} \sum_{k=1}^{K} \left( a^{4k} - 2a^{2k} + 1 \right).$$

Substituting $K = \log_a n + 1$ (so that $a^{2K} = a^2 n^2$ and $a^{4K} = a^4 n^4$) yields

$$C^{\text{VAR}} = \frac{1}{(a^2 - 1)^2} \left[ \frac{a^4(n^4 - 1)}{a^4 - 1} - \frac{2a^2(a^2 n^2 - 1)}{a^2 - 1} + (\log_a n + 1) \right].$$

The asymptotic time complexity is governed by the dominant term:

$$C^{\text{VAR}} \sim \frac{a^8}{(a^4 - 1)(a^2 - 1)^2} n^4 = \mathcal{O}(n^4).$$

**Diffusion.** Assume a self-attention DiT where each diffusion step uses the fixed largest resolution $n \times n$, i.e., $n^2$ tokens. So, a single step costs

$$C_1^{\text{Diff}} = (n^2)^2 = n^4.$$

With the same number of forward steps as VAR, namely $K = \log_a n + 1$, we get

$$C^{\text{Diff}} = \sum_{k=1}^{K} n^4 = (\log_a n + 1) n^4 = \mathcal{O}(n^4 \log n).$$

**Comparison.** From the above,

$$\frac{C^{\text{Diff}}}{C^{\text{VAR}}} \sim \frac{(\log_a n) \, n^4}{\frac{(a^4-1)(a^2-1)^2}{a^8} n^4} = \frac{a^8}{(a^4 - 1)(a^2 - 1)^2} \log_a n = \mathcal{O}(\log n).$$

That is, with the same number of forward passes, VAR totals $\mathcal{O}(n^4)$ complexity while diffusion totals $\mathcal{O}(n^4 \log n)$, yielding a $\mathcal{O}(\log n)$ speedup for VAR.

## D  ADDITIONAL ABLATIONS

In this section, we provide additional ablations and analysis of RestoreVAR and its components.

### D.1  PERFORMANCE ANALYSIS: APE VS. ROPE

As discussed in Sec. 3.3, we replace the absolute position embeddings (APE) used in VAR (Tian et al., 2024) with Rotary Positional Embeddings (Su et al., 2024) (RoPE). We found that using RoPE yields better performance when scaling the resolution from $256 \times 256$ to $512 \times 512$.

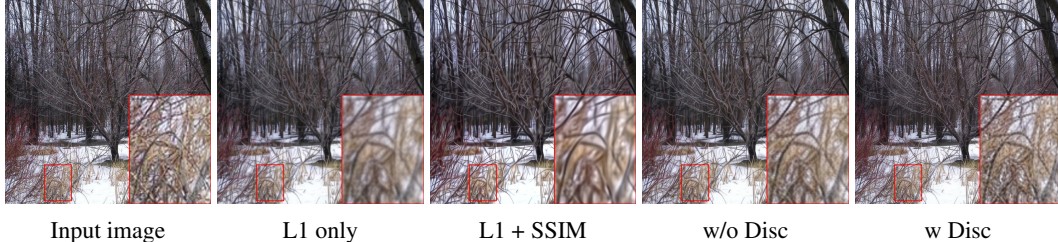

| Input image | L1 only | L1 + SSIM | w/o Disc | w Disc |
|---|---|---|---|---|

Figure 12: Image reconstructed by VAE decoders fine-tuned on continuous latents with various objective functions.

To demonstrate this, we conducted an ablation where both APE and RoPE-based variants were fine-tuned at $512 \times 512$ resolution for 10 epochs. As shown in Fig. 11, the RoPE-based model achieves higher validation accuracy compared to the APE-based model, indicating its effectiveness.

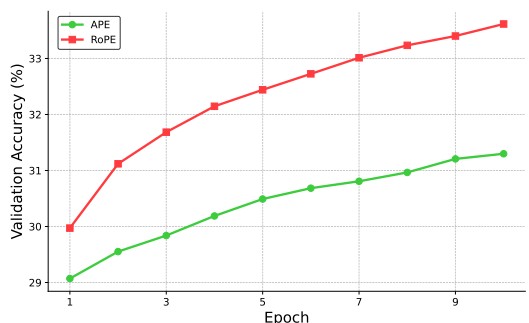

Figure 11: Validation accuracy comparison of APE and RoPE-based fine-tuning at $512 \times 512$ resolution. RoPE demonstrates better performance.

### D.2 ADDITIONAL VAE LOSS ABLATIONS

We now provide ablations on fine-tuning the VAE decoder without SSIM and perceptual losses. Fig. 12 illustrates that using only L1 loss produces blurry results, adding the SSIM losses enhances structural details and incorporating the perceptual loss improves sharpness while preserving the structure (w/o Disc column). Incorporating the discriminator enhances sharpness even further.

### D.3 ADDITIONAL SCALE-SPACE ANALYSIS

We provide an analysis similar to Fig. 2 for real-world degradations using the POLED and REVIDE datasets. The results in Fig. 13 indicate that the scale-space decomposition of VAR tends to capture real-world degradations in coarse scales while preserving scene-level details in finer scales. This suggests that the scale-space behavior observed in the synthetic examples of Fig. 2 also extends to real-world degradations.

### D.4 ADDITIONAL LRT ANALYSIS

In this section, we analyze if the LRT module can perform restoration without relying on the VAR transformer. Toward this aim, we conduct two experiments:

$$\hat{f}_{\text{cont}} = f_{\text{quant}}^{\text{deg}} + \text{LRT}(f_{\text{quant}}^{\text{deg}}, z),$$

and

$$\hat{f}_{\text{cont}} = f_{\text{cont}}^{\text{deg}} + \text{LRT}(f_{\text{cont}}^{\text{deg}}, z),$$

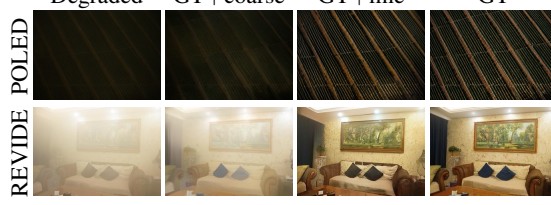

Figure 13: Our analysis that VAR captures degradations in early scales (coarse) and scene-level details in later scales (fine) also tends to hold for real degradations from POLED and REVIDE datasets. GT+coarse replaces early GT scales with degraded ones, while GT+fine replaces the late GT scales.

where $f_{\text{quant}}^{\text{deg}}$ and $f_{\text{cont}}^{\text{deg}}$ denote the quantized and continuous VAE latents obtained directly from the degraded input. Note that we are passing $z$ as input to the LRT since the LRT is trained to utilize $z$ to predict a meaningful residual. Zeroing out $z$ leads to significant artifacts as we are giving incorrect guidance to the LRT. Fig. 14 provides

| Input | $f_{\text{quant}}^{\text{deg}} + \text{LRT}(f_{\text{quant}}^{\text{deg}}, z)$ | $f_{\text{cont}}^{\text{deg}} + \text{LRT}(f_{\text{cont}}^{\text{deg}}, z)$ | GT |
|---|---|---|---|

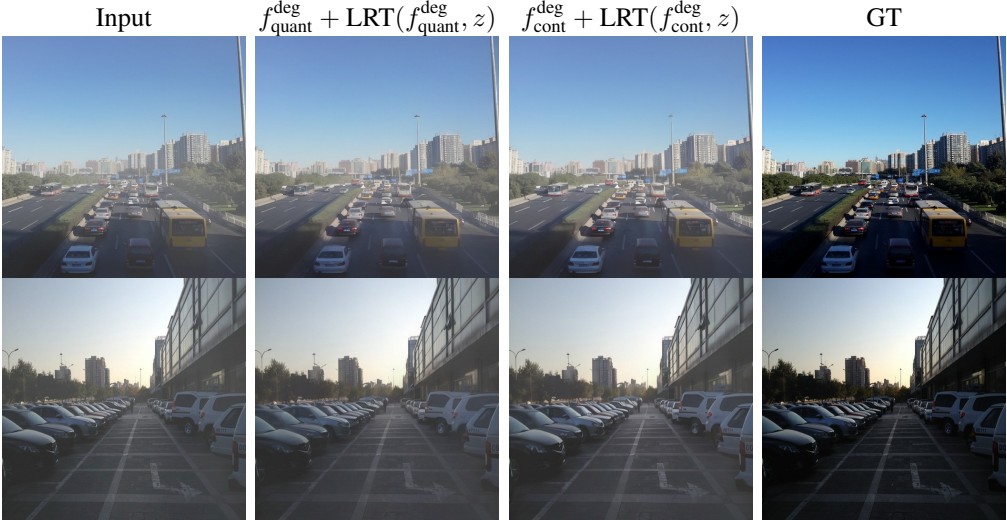

Figure 14: Qualitative analysis of the LRT module operating directly on the degraded quantized and continuous latents. The prediction of the LRT remains hazy, indicating that the LRT cannot restore images without the VAR transformer.

the results for these experiments. Despite having access to the restored latent's representation $z$, the LRT is unable to recover the clean latent (and so is the VAE decoder). This demonstrates that the LRT and the VAE decoder cannot independently perform restoration. These experiments reaffirm that restoration is carried out mainly by the VAR transformer, while the LRT and decoder serve to enhance fidelity by operating on VAR's restored latent representation.

## E  ADDITIONAL QUANTITATIVE RESULTS

In this section, we provide additional quantitative analysis of the results in Table 1, Table 2 and Table 3.

### E.1  USER STUDY

We now provide rigorous statistical analysis for the user study, by computing paired differences between RestoreVAR and each method across all 36 responses ($\Delta = \text{RestoreVAR} - \text{Method}$), and report $95\%$ confidence intervals (CI), paired t-tests, and Wilcoxon signed-rank tests. The results, summarized in Table 7, show that RestoreVAR significantly outperforms all methods: every mean improvement $\Delta$ is positive and the $95\%$ CIs do not cross zero. Furthermore, the t-tests and Wilcoxon signed-rank tests report $p < 10^{-4}$. These results demonstrate that the advantages of RestoreVAR over non-generative methods are statistically significant across user study responses.

We also provide additional details for the user study methodology. We selected 50 images for evaluation and collected 36 independent responses. Participants evaluated each result on a five-point scale for (i) restoration performance,

Welcome to our survey! In the next question, a google drive link is provided. When you open the link, you would find a set of images. Each image consists of 2 images stitched side by side. The image on the left is that of a scene degraded by rain, haze or low-light, etc (input image). The image on the right is the same scene with degradations removed (restored image). Based on the visual difference, please assess the restored image on 3 metrics:

1. Scene consistency: How similar is the scene content (fine details such as text) between the input and restored images on a scale on 1-5 (1: not consistent, 5: very consistent)?

2. Restoration performance: How much degradation was removed on a scale on 1-5 (1: very less degradation removed, 5: lot of degradation removed)?

3. Overall quality: How good is the visual quality of the restored image on a scale on 1-5 (1: poor quality, 5: high quality)?

Every image in the drive folder has a number at the beginning of it. These questions need to be answered by entering the score for each metric for each image number (given in the question).

Figure 15: Screenshot of the user study instruction page.

Please fill out the table for each image with number 'x' at the beginning (separated by '_'), where 'x' is given in the first column of the below table.

| | Scene consistency | Restoration performance | Overall quality |
|---|---|---|---|
| 2 | ⌄ | ⌄ | ⌄ |
| 3 | ⌄ | ⌄ | ⌄ |
| 4 | ⌄ | ⌄ | ⌄ |
| 5 | ⌄ | ⌄ | ⌄ |
| 6 | ⌄ | ⌄ | ⌄ |
| 7 | ⌄ | ⌄ | ⌄ |

Figure 16: Screenshot of the user study data collection page.

Table 7: Statistical analysis of user study results using $95\%$ confidence intervals (CIs), paired t-test and Wilcoxon signed-rank test. $\Delta$ denotes the mean paired difference between RestoreVAR and each method.

| Method | $\Delta$ | 95% CI | $p$ (t-test) | $p$ (Wilcoxon) |
|---|---|---|---|---|
| AutoDIR | 0.694 | $\pm 0.257$ | $6.49 \times 10^{-6}$ | $3.06 \times 10^{-5}$ |
| AWRaCLe | 2.028 | $\pm 0.245$ | $7.15 \times 10^{-18}$ | $2.91 \times 10^{-11}$ |
| PromptIR | 2.250 | $\pm 0.221$ | $1.12 \times 10^{-20}$ | $2.91 \times 10^{-11}$ |
| DCPT | 1.944 | $\pm 0.262$ | $2.15 \times 10^{-16}$ | $2.91 \times 10^{-11}$ |
| InstructIR | 1.417 | $\pm 0.293$ | $3.28 \times 10^{-11}$ | $1.66 \times 10^{-6}$ |

(ii) overall quality, and (iii) scene consistency.

The survey was conducted using Qualtrics (Provo, UT). Screenshots of the survey are given in Figs. 15 and 16. Fig. 15 shows the introduction page of the user study. The user would be provided with a randomly sampled google drive link consisting of restored outputs of different methods, where each method is assigned a number. The user needs to evaluate each of these images on the afore-mentioned criteria and enter the scores in the table shown in Fig. 16.

### E.2    GENERALIZATION METRICS

We additionally perform statistical significance analysis for our quantitative results in Table 2. Specifically, we compute per-image paired differences between RestoreVAR and each non-generative method on MUSIQ and CLIPIQA and apply dataset-weighted statistical tests to ensure that each dataset contributes equally to the global statistic. As shown in Table 8, RestoreVAR consistently achieves positive improvements with tight CIs and extremely small $p$ values. Once again, these results confirm that RestoreVAR achieves statistically significant gains in perceptual image-quality metrics over non-generative methods.

### E.3    MORE PERCEPTUAL QUALITY EVALUATIONS

We provide no-reference metrics MUSIQ and CLIPIQA which gauge perceptual quality for the test sets in Table 1 where there is a significant gap between non-generative methods and RestoreVAR on pixel-level metrics. From the MUSIQ/CLIPIQA scores provided in Table 9, it can be observed that the perceptual quality of RestoreVAR is on-par or even higher than those of non-generative methods, indicating perceptually comparable results. However, due to lower pixel-level fidelity than non-generative approaches, RestoreVAR scores lower on pixel-level metrics.

## F    ADDITIONAL ARCHITECTURAL DETAILS

We now provide additional architectural details for the RestoreVAR framework. We first describe the details for the RestoreVAR transformer, followed by the Latent Refiner Transformer (LRT).

**RestoreVAR Transformer.**   We adopted the VAR model with a transformer depth of 16, i.e., the architecture consists of 16 transformer blocks. The structure of each block is illustrated in

Table 8: Statistical analysis for results from Table 2 using $95\%$ confidence intervals (CIs), paired t-test and Wilcoxon signed-rank test. $\Delta$ denotes the weighted mean paired difference between RestoreVAR and each method.

| Method | MUSIQ | | | CLIPIQA | | |
|---|---|---|---|---|---|---|
| | $\Delta$ | 95% CI | $p$-value | $\Delta$ | 95% CI | $p$-value |
| AWRaCLe | 5.084 | $\pm 0.554$ | $2.5 \times 10^{-55}$ | 0.0536 | $\pm 0.0080$ | $8.5 \times 10^{-34}$ |
| DCPT | 5.206 | $\pm 0.613$ | $3.0 \times 10^{-49}$ | 0.0166 | $\pm 0.0079$ | $3.9 \times 10^{-5}$ |
| DFPIR | 8.276 | $\pm 0.757$ | $2.5 \times 10^{-71}$ | 0.0733 | $\pm 0.0093$ | $3.6 \times 10^{-44}$ |
| InstructIR | 6.984 | $\pm 0.902$ | $1.3 \times 10^{-42}$ | 0.0495 | $\pm 0.0065$ | $1.8 \times 10^{-41}$ |
| PromptIR | 7.002 | $\pm 0.648$ | $3.3 \times 10^{-70}$ | 0.0395 | $\pm 0.0068$ | $8.4 \times 10^{-27}$ |

Table 9: Comparison of perceptual quality of RestoreVAR and non-generative methods on RESIDE, Snow100k, Rain13K, LOLv1 and GoPro datasets using MUSIQ and CLIPIQA metrics.

| Method | GoPro | | LOLv1 | | RESIDE | | Rain13K | | Snow100K | | Average | |
|---|---|---|---|---|---|---|---|---|---|---|---|---|
| | MUSIQ↑ | CLIPIQA↑ | MUSIQ↑ | CLIPIQA↑ | MUSIQ↑ | CLIPIQA↑ | MUSIQ↑ | CLIPIQA↑ | MUSIQ↑ | CLIPIQA↑ | MUSIQ↑ | CLIPIQA↑ |
| PromptIR | 35.87 | 0.155 | 50.24 | 0.306 | 66.54 | 0.434 | 64.63 | 0.471 | 63.29 | 0.417 | 56.11 | 0.357 |
| InstructIR | 53.06 | 0.251 | 67.55 | 0.364 | 67.59 | 0.449 | 66.51 | 0.504 | – | – | 63.68 | 0.392 |
| AWRaCLe | 44.953 | 0.173 | 63.198 | 0.459 | 61.938 | 0.346 | 64.683 | 0.598 | 62.334 | 0.512 | 59.421 | 0.418 |
| DCPT | 46.12 | 0.180 | 67.73 | 0.404 | 66.46 | 0.432 | 62.63 | 0.437 | – | – | 60.73 | 0.363 |
| DFPIR | 48.03 | 0.190 | 66.77 | 0.409 | 66.55 | 0.423 | 62.28 | 0.432 | – | – | 60.91 | 0.364 |
| **RestoreVAR** | 55.45 | 0.231 | 71.47 | 0.396 | 66.45 | 0.481 | 63.83 | 0.516 | 63.56 | 0.482 | 64.15 | 0.421 |

Fig. 17(a). The embedding dimension was set to $1024$, and the number of attention heads used was $16$. Furthermore, the transformer predicted discrete latents at the following spatial resolutions in the latent space: $1 \times 1$, $2 \times 2$, $3 \times 3$, $4 \times 4$, $6 \times 6$, $9 \times 9$, $13 \times 13$, $18 \times 18$, $24 \times 24$, and $32 \times 32$. The start-of-sequence (SOS) token is constructed by augmenting the class embedding with the mean value (along spatial dimensions) of the features obtained after a learnable projection applied on $f_{\text{cont}}^{\text{deg}}$. Specifically,

$$\text{SOS} = \text{class}_{\text{emb}} + g_{\text{sos}} \times \text{Mean}(\text{Proj}(f_{\text{cont}}^{\text{deg}}), \text{SOS} \in \mathbb{R}^{1 \times C}.$$

Here, $\text{class}_{\text{emb}}$ is the class token embedding and $g_{\text{sos}}$ is initialized as $0$ for gradual incorporation of degradation conditioning. Other notations follow Sec. 3.

**Latent Refiner Transformer.** The LRT follows a similar structure for the blocks as the RestoreVAR transformer, as shown in Fig. 17(b). It was configured with a depth of 12, six attention heads, and an embedding dimension of $384$.

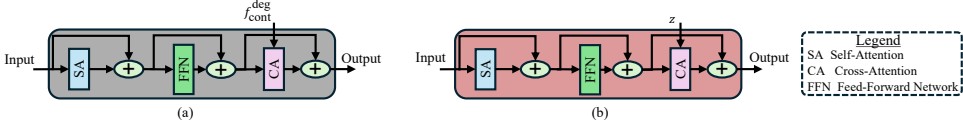

Figure 17: Illustration of a transformer block in (a) RestoreVAR transformer and (b) Latent Refiner Transformer.

# G RUNTIME AND PARAMETER BREAKDOWN

In this section, we provide a breakdown of the runtime and parameter count for the following components of the RestoreVAR framework: the VAE, the RestoreVAR transformer, and the Latent Refiner Transformer (LRT). This analysis provides insights into the distribution of the computational cost across the pipeline. As shown in Table 10, the majority of inference time is taken by the autoregressive RestoreVAR transformer.

| Component | VAE | Transformer | Refiner |
|---|---|---|---|
| **Time (s)** | 0.0086 | 0.1863 | 0.0061 |
| **Parameters (M)** | 108.95 | 273.98 | 22.97 |

Table 10: Compute time and parameter count breakdown for each component of RestoreVAR. VAE time includes both encoding and decoding.

## H    ADDITIONAL VISUAL RESULTS

We now present additional visualizations for some of
the ablations discussed in Sec. 4.4, along with more qualitative comparisons across methods.

### H.1    CONTINUOUS VS. DISCRETE CONDITIONING

As shown in Sec. 4.4, conditioning RestoreVAR on the continuous latent of the degraded image yields
significantly better performance compared to using the quantized or discrete latent. Fig. 18 further
illustrates this using visual comparisons between the model trained with discrete and continuous
conditioning. The model trained with discrete conditioning exhibits noticeably more hallucinations
than the one trained with continuous conditioning.

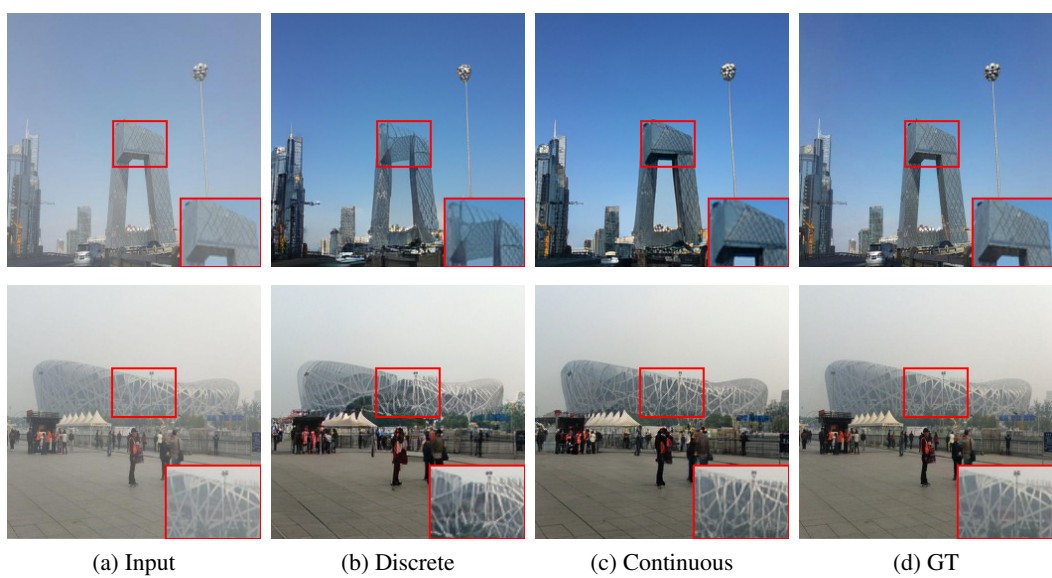

| (a) Input | (b) Discrete | (c) Continuous | (d) GT |

Figure 18: Qualitative comparisons of RestoreVAR under discrete vs. continuous conditioning.
RestoreVAR with discrete conditioning exhibits more hallucinations than the variant with continuous
conditioning.

### H.2    QUALITATIVE COMPARISONS OF VAE DECODERS

As mentioned in Sec. 3.3.2, our
fine-tuned VQ-VAE decoder achieves
superior reconstruction performance
compared to the decoders of VAR and
HART. Fig. 19, provides qualitative
results to illustrate the same. Our de-
coder produces the best reconstruc-
tion.

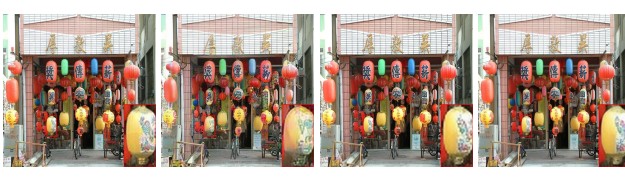

(a) Input    (b) VAR    (c) HART    (d) Ours

Figure 19: Qualitative comparisons of the input reconstructed
using VAR (Tian et al., 2024), HART (Tang et al., 2024) and
Our VAE decoder. Our result has minimal distortions.

### H.3    VISUAL COMPARISON OF REFINER VARIANTS

In Sec. 4.4, we demonstrated that our proposed Latent Refiner Transformer (LRT) achieves the best
performance compared to using no refiner, a refiner without last-block conditioning, and HART (Tang
et al., 2024)'s diffusion-based refiner. Quantitative results, reported in Table 4, included mean PSNR
and SSIM scores on the RESIDE (Li et al., 2019) test set. Fig. 20 presents qualitative comparisons
for these configurations. It can be observed that our LRT preserves fine details more effectively than
the other variants.

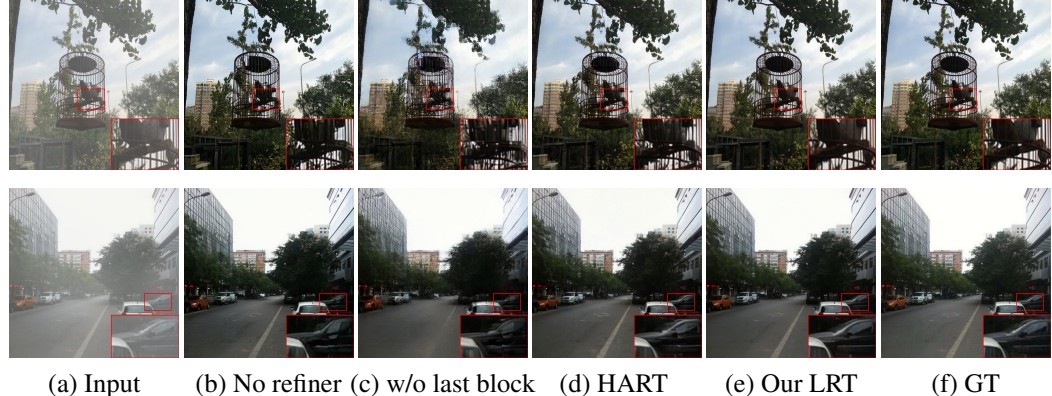

| (a) Input | (b) No refiner | (c) w/o last block | (d) HART | (e) Our LRT | (f) GT |

Figure 20: Qualitative results for the ablation on latent refiner configurations. Our proposed LRT preserves fine details better than the other configurations.

### H.4 ADDITIONAL QUALITATIVE COMPARISONS

In this section, we provide additional qualitative comparisons of RestoreVAR with state-of-the-art LDM-based and non-generative all-in-one image restoration (AiOR) methods. Fig. 21 presents results from RestoreVAR alongside Diff-Plugin(Liu et al., 2024), AutoDIR (Jiang et al., 2023), and PixWizard (Lin et al., 2024) on the RESIDE (Li et al., 2019), Snow100k (Liu et al., 2018), Rain13K (Zamir et al., 2021), LOLv1 (Wei et al., 2018), and GoPro (Nah et al., 2017) datasets. RestoreVAR consistently produces outputs that are more semantically aligned with the ground truth (see zoomed-in patches).

Fig. 22 provides comparisons with non-generative methods—PromptIR(Potlapalli et al., 2024), InstructIR (Conde et al., 2025), AWRaCLe (Rajagopalan & Patel, 2024), DCPT (Hu et al., 2025) and DFPIR (Tian et al., 2025)—for real-world, mixed and unseen degradation generalization on LHP (Guo et al., 2023), REVIDE (Zhang et al., 2021b), TOLED (Zhou et al., 2021), POLED (Zhou et al., 2021), CDD (Guo et al., 2024) and LOLBlur (Zhou et al., 2022) datasets. RestoreVAR generates sharper, more realistic outputs with fewer artifacts than non-generative models. For instance, for the TOLED and POLED cases, RestoreVAR outputs are visibly sharper than non-generative methods. Similarly, the results of RestoreVAR are superior in the case of mixed degradations.

## I MORE DETAILS ABOUT AUTODIR COMPARISON

As mentioned in the main paper, comparisons with AutoDIR (Jiang et al., 2023) were conducted without its structure correction module (SCM). AutoDIR consists of a latent diffusion model (LDM) for initial restoration, followed by an SCM which is a non-generative post-processing network. The intuition behind this approach is that the SCM predicts a residual based on the degraded input image and the restored output of the LDM, to correct the VAE-induced distortions. In short,

$$I_{\text{result}} = I_{\text{sd}} + \mathcal{F}\left([I_{\text{sd}}, I_{\text{deg}}]\right),$$

where $I_{\text{sd}}$ is the restored output from the LDM, $I_{\text{deg}}$ is the original degraded input image, and $\mathcal{F}(\cdot)$ denotes the SCM which operates on the concatenated inputs $[I_{\text{sd}}, I_{\text{deg}}]$. However, we found that instead of slightly modulating the structural details in $I_{\text{sd}}$, the SCM behaves like a separate non-generative restoration model which directly restores $I_{\text{deg}}$. To show this, we evaluated AutoDIR with the SCM on the RESIDE (Li et al., 2019) dataset for two cases: (1) using $I_{\text{sd}}$ as the actual LDM output and (2) setting $I_{\text{sd}} = 0$, effectively removing any structural information from the LDM. If the SCM were functioning as a corrective module, performance in the second case should deteriorate significantly. However, we found that the SCM was able to independently restore the degraded input in the second case, as shown in Fig. 24. This suggests that the SCM largely ignores the LDM output and instead performs direct restoration on $I_{\text{deg}}$, thereby behaving as a non-generative restoration network. Therefore, to ensure a fair comparison with other generative models, we evaluated only the LDM output of AutoDIR.

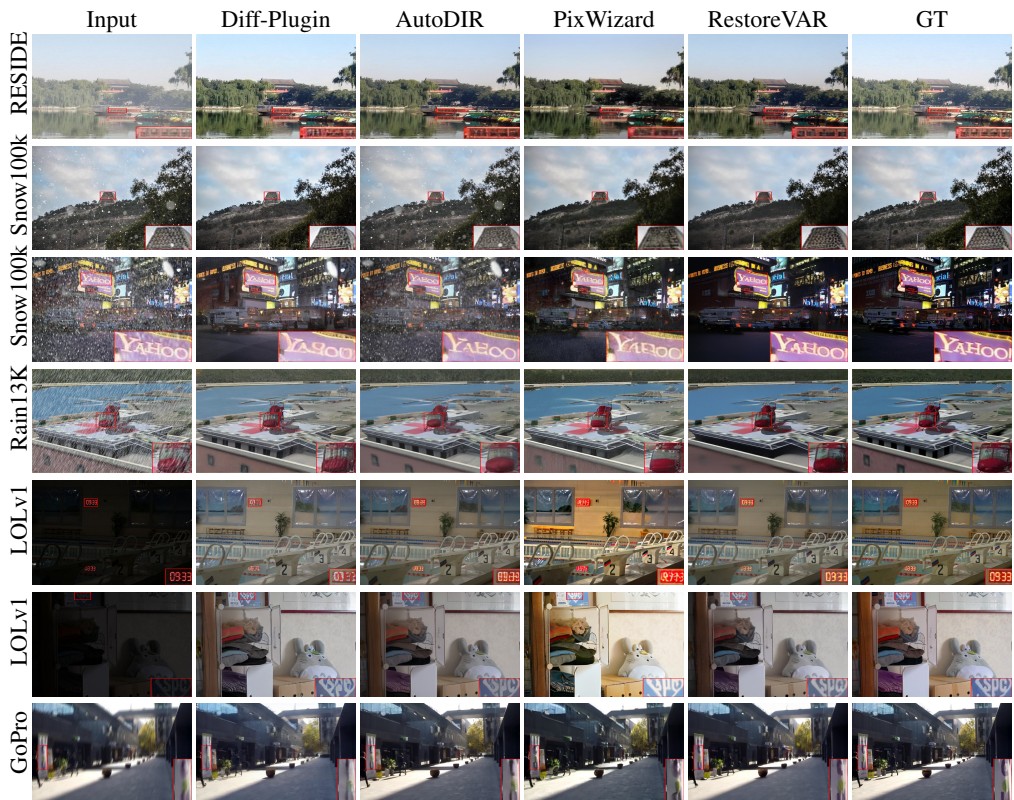

Figure 21: Additional qualitative comparisons of RestoreVAR with LDM-based AiOR approaches. RestoreVAR consistently preserves fine details more effectively than the LDM-based methods.

## J LIMITATIONS AND SCOPE FOR FUTURE WORK

Despite the strengths of RestoreVAR, there remains scope for improvement. First, its performance is inherently constrained by the latent refiner transformer (LRT) and the VAE decoder. While the LRT significantly improves results over using no refiner, it does not reach the upper bound set by directly decoding from ground-truth continuous latents. Exploring improved VQVAE and refiner architectures could help address this. Another promising direction is to employ our non-generative LRT in fully generative VAR models, given its strong performance for AiOR. Finally, future work can investigate how the performance of RestoreVAR scales with larger VAR backbones.

Additionally, we found that RestoreVAR does not perform well when the input degradation is extremely severe. Fig. 25 shows the result for a very dark low-light sample from the SID (Chen et al., 2018a) dataset. The restored output looks inferior to the GT due to the very severe nature of the degradation.

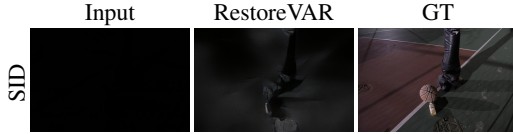

Figure 25: RestoreVAR does not perform well when the input degradation is very severe.

## K LLM USAGE

LLM was used only for polishing writing in parts of the main paper and supplementary.

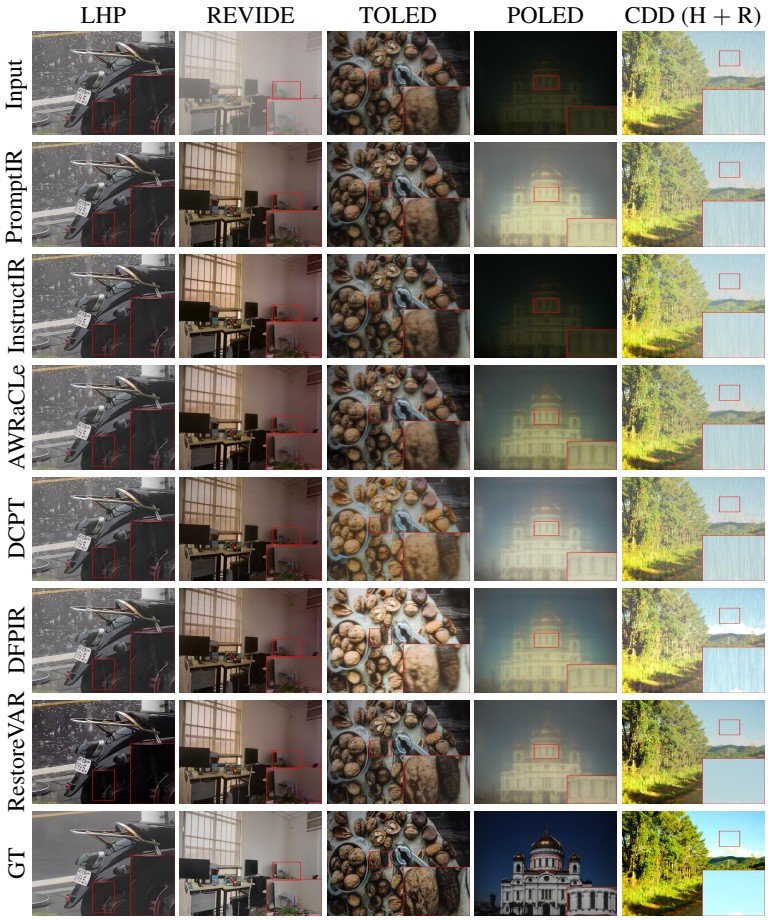

Figure 22: Additional qualitative comparisons of RestoreVAR with non-generative methods on real-world, unseen and mixed degradations. RestoreVAR achieves better results, highlighting its superior generalization.

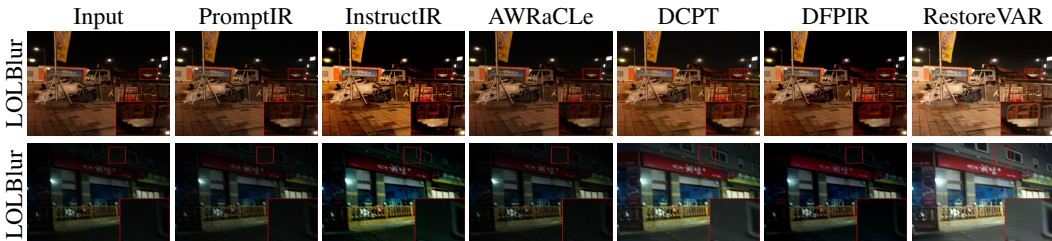

Figure 23: Qualitative comparisons with non-generative methods on samples from the real mixed-degradation dataset LOLBlur.

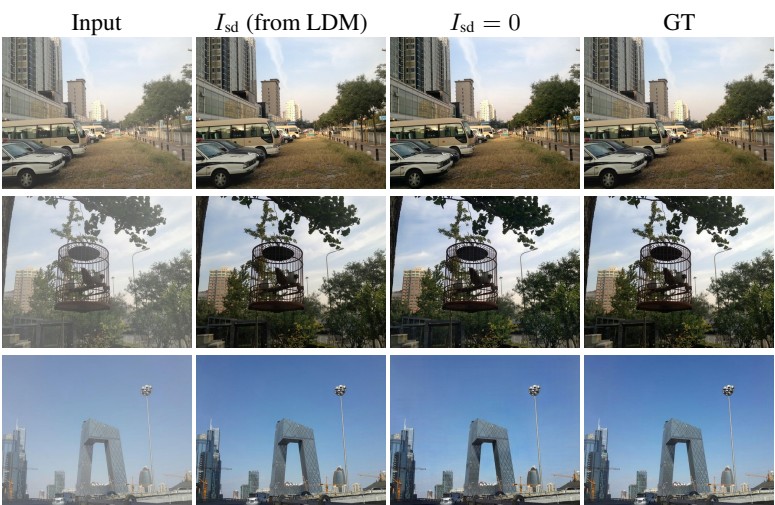

Figure 24: Illustration of the behavior of AutoDIR's (Jiang et al., 2023) structure correction module (SCM). The second column shows outputs when the SCM is applied to the LDM output $I_{sd}$, while the third column shows results when $I_{sd}$ is set to zero. Despite no structural information (third column), the SCM still restores the image, indicating that it functions as a separate non-generative restoration model.

