# OpenReview forum: "RestoreVAR: Visual Autoregressive Generation for All-in-One Image Restoration"
_ICLR.cc/2026/Conference — ICLR 2026 Poster_

### Official Review · Reviewer_HSQV · 2025-10-23

**Soundness:** 3
**Presentation:** 3
**Contribution:** 2
**Rating:** 4
**Confidence:** 2

**Summary:**

This paper proposes RestoreVAR, an All-in-One Image Restoration (AiOR) model that is built on a pre-trained Visual Auto-Regressive (VAR) model. This method leverages the multi-scale prior knowledge in the VAR model and is thus well-suited for the AiOR task. To achieve this, the authors introduce a cross-attention mechanism for conditioning the pre-trained VAR model on the degradation images. Finally, to mitigate the information loss due to the vector quantization operations, a transformer-based post-processing model is introduced to refine the generated tokens. Similarly, the VAE decoder is fine-tuned to further improve the reconstruction quality. Together, RestoreVAR achieves state-of-the-art performances as the diffusion-based generative-IR models while being 10 times faster at inference.

**Strengths:**

- The method’s design is intuitive and reasonable. The proposed method is tailored for the VAR, utilizing its efficient architecture while offering good sample quality.
- The motivation is well presented and analyzed. In Section 3.1 and 3.2, the authors presented their observations, where the multi-scale structure of the VAR model effectively simplifies the AiOR task to the generation of low resolution features.
- The manuscript is overall well-structured and easy reading.

**Weaknesses:**

- As mentioned in Section 3, the generative-IR methods are known to suffer from hallucinations that introduce artifacts in the results and affect metrics like PSNR and SSIM. Meanwhile, the generative nature of these methods offer improved image quality than their non-generative counterparts. This balance is also known as perception-distortion tradeoff. While RestoreVAR demonstrates advantages in PSNR and SSIM over other generative methods, they are not quantitatively compared on the image quality metrics like MUSIQ and CLIP-IQA. Specifically, Table 3 only presents a comparison in such metrics between RestoreVAR, a generative method, and other non-generative methods. I believe more discussions on the perception-distortion tradeoff will better demonstrate the advantages of RestoreVAR over other generative-IR methods.
- Although the efficiency advantage of RestoreVAR is well established, the advantage in terms of generation quality of a VAR backbone is unclear. I believe Table 4 shall also include image quality metrics. Without LRT, the performance of RestoreVAR on PSNR and SSIM metrics appears to be the same level as other generative-IR models (21.71/0.690 v.s. 21.28/0.738). If LRT compromised image quality, the improvements in generation quality of RestoreVAR will be limited, especially considering other generative-IR methods were not evaluated on a fine-tuned VAE decoder.

**Questions:**

- To help us better understand Weakness 1, could authors provide a set of ablation studies on the performance gains on the predicted higher resolution index maps? Specifically, a quantitative comparison between images with high-res index maps generated from ground-truth low-res index maps and from scratch generation. In other words, comparisons between [GT-low-res, Gen-high-res (from GT-low-res)] and [GT-low-res, Gen-high-res (from scratch)]? Additionally you could also provide another set of [Gen-low-res, Gen-high-res (from GT-low-res)] and [Gen-low-res, Gen-high-res (from scratch)], to better demonstrate the difference.
- Given the limitations in Weakness 1 and 2, could authors discuss more on the evidence that how to justify RestoreVAR to be claimed as state-of-the-art, instead of another point on the perception-distortion plane?

---

> ### Author Response · Authors · 2025-11-22
> **Response to reviewer HSQV (W1, W2)**
>
> We thank the reviewer for their valuable feedback. We now address the concerns raised.
>
> **W1:  Other generative methods are not quantitatively compared on MUSIQ and CLIP-IQA. More discussions on the perception-distortion tradeoff will better demonstrate the advantages of RestoreVAR over other generative-IR methods.**
>
> We have now included MUSIQ and CLIPIQA scores for LDM-based approaches in Table 2 (or see Table R1). These results show that RestoreVAR achieves perceptual quality comparable to that of LDM-based models, while significantly outperforming non-generative methods. From the perspective of the perception–distortion tradeoff, distortions can arise either from *residual degradations* left in the restored image or from *model-induced artifacts* such as hallucinations. From Table 2 and Fig. 6, non-generative methods often produce results of poor perceptual quality that contain distortions characterized by residual degradations. LDM-based generative approaches produce results of high perceptual quality but suffer from model-induced distortions that compromise pixel-level fidelity. RestoreVAR achieves comparable perceptual quality and restoration performance to those of LDM-based methods with superior pixel-level fidelity, and thus, fewer distortions. Although its fidelity does not yet reach that of non-generative approaches, RestoreVAR reduces model-induced distortions relative to the diffusion-based generative methods while simultaneously preserving the strong generalization capabilities of LDM-based approaches with faster inference speed.
>
> Key points from this discussion are included in lines 455-469 of the revised paper.
>
> **Table R1: Quantitative comparisons of RestoreVAR against state-of-the-art non-generative and generative approaches on real-world, unseen, and mixed degradations.**
>
> | **Method**|**LHP** MUSIQ↑|LHP CLIPIQA↑|**REVIDE** MUSIQ↑|REVIDE CLIPIQA↑|**TOLED** MUSIQ↑|TOLED CLIPIQA↑|**POLED** MUSIQ↑|POLED CLIPIQA↑|**LOLBlur** MUSIQ↑|LOLBlur CLIPIQA↑|**CDD** MUSIQ↑|CDD CLIPIQA↑| **Average** MUSIQ↑|Average CLIPIQA↑|
> |-|-|-|-|-|-|-|-|-|-|-|-|-|-|-|
> | PromptIR | 56.780 | 0.366 | 61.191 | 0.459 | 43.218 | 0.281 | 34.536 | 0.303 | 33.693 | 0.166 | 65.895 | 0.483 | 49.219 | 0.343 |
> | InstructIR | 58.269 | 0.359 | 63.116 | 0.416 | 44.985 | 0.298 | 23.317 | 0.241 | 40.221 | 0.202 | 65.491 | 0.482 | 49.900 | 0.333 |
> | AWRaCLe | 57.889 | 0.333 | 59.287 | 0.368 | 44.670 | 0.285 | 40.533 | 0.332 | 38.186 | 0.171 | 66.253 | 0.484 | 51.470 | 0.329 |
> | DCPT | 58.044 | 0.372 | 60.011 | 0.446 | 44.062 | 0.314 | 38.138 | 0.345 | 37.393 | 0.175 | 68.440 | 0.544 | 51.681 | 0.366 |
> | DFPIR | 56.483 | 0.330 | 61.009 | 0.450 | 43.820 | 0.276 | 35.668 | 0.289 | 36.277 | 0.163 | 54.408 | 0.349 | 47.611 | 0.310 |
> | DiffPlugin | 57.351 | 0.420 | 63.483 | 0.406 | 46.219 | 0.311 | 35.086 | 0.407 | 40.054 | 0.212 | 68.791 | 0.578 | 51.831 | 0.389 |
> | AutoDir | 58.085 | 0.380 | 63.918 | 0.416 | 54.585 | 0.341 | 48.796 | 0.380 | 46.642 | 0.225 | 68.575 | 0.571 | 56.767 | 0.386 |
> | PixWizard | 58.600 | 0.411 | 68.487 | 0.420 | 45.804 | 0.329 | 44.305 | 0.348 | 50.708 | 0.268 | 68.409 | 0.589 | 56.052 | 0.394 |
> | RestoreVAR | 57.662 | 0.414 | 63.562 | 0.483 | 52.374 | 0.338 | 48.118 | 0.276 | 46.644 | 0.214 | 68.941 | 0.572 | 56.217 | 0.383 |
>
> **W2: Advantage in terms of generation quality of a VAR backbone is unclear. Provide image quality metrics in Table 4 as well to check if LRT compromised image quality.**
>
> The VAR backbone is responsible for removing degradations, which in-turn produces perceptually appealing images. For instance, in Fig. 4 and Fig. 20, the output of VAR without the LRT and vanilla VQVAE decoder (column "discrete" in Fig. 4 and column (b) in Fig. 20) already exhibits strong degradation mitigation. To assess if the LRT causes loss of perceptual quality, we calculated MUSIQ and CLIPIQA scores (higher is better) with and without the LRT. Using no LRT yields a MUSIQ/CLIPIQA score of $66.56/0.472$ while incorporating the LRT results in a score of $66.45/0.481$. The scores are nearly identical indicating negligible impact on perceptual quality.
>
> The perceptual quality metrics with and without the LRT are included in lines 500-507 of the revised paper.

---

> ### Author Response · Authors · 2025-11-22
> **Response to reviewer HSQV (Q1, Q2)**
>
> **Q1: To better understand Weakness 1, could authors provide a quantitative comparisons between [GT-low-res, Gen-high-res (from GT-low-res)] and [GT-low-res, Gen-high-res (from scratch)]? Additionally you could also provide another set of [Gen-low-res, Gen-high-res (from GT-low-res)] and [Gen-low-res, Gen-high-res (from scratch)], to better demonstrate the difference.**
>
> We have provided the comparisons for experiments requested by the reviewer in Table R2. From the table, the RestoreVAR (Gen-LR $\rightarrow$ Gen-HR (from Gen-LR)) prediction yields best results. This is because, all scales are generated by the model itself rather than using the GT scales resulting in model-predicted scales that are better aligned to its past predictions.
>
> **Table R2: Average PSNR (dB), SSIM, and LPIPS across RESIDE, Snow100K, Rain13K, LOLv1, and GoPro datasets for the analysis requested by the reviewer. The last row is analogous to RestoreVAR.**
>
> | **Setting** | **PSNR↑** | **SSIM↑** | **LPIPS↓** |
> |-------------|-----------|-----------|------------|
> | GT-LR → Gen-HR (from GT-LR)      | 23.78 | 0.7371 | 0.1390 |
> | GT-LR → Gen-HR (from scratch)    | 23.10 | 0.7383 | 0.1463 |
> | Gen-LR → Gen-HR (from GT-LR)     | 21.78 | 0.7175 | 0.1694 |
> | Gen-LR → Gen-HR (from Gen-LR)    | 23.67 | 0.7517 | 0.1345 |
>
>
> **Q2: Given the limitations in Weakness 1 and 2, could authors discuss more on the evidence that how to justify RestoreVAR to be claimed as state-of-the-art, instead of another point on the perception-distortion plane?**
>
> We have addressed Weaknesses 1 and 2 by (i) providing perceptual metrics (MUSIQ and CLIPIQA) for LDM-based generative models in Table 2 (or Table R1) to discuss perception-distortion tradeoffs, and (ii) quantifying the perceptual impact of the LRT. From Table 2, RestoreVAR obtains perceptual quality comparable to that of diffusion-based generative AiOR models while offering substantially higher pixel-level fidelity and significantly faster inference. Conversely, non-generative approaches exhibit high fidelity but suffer from poor perceptual quality and limited generalization, especially in the presence of unseen or mixed degradations. Subsequently, we showed that the LRT does not compromise perceptual quality: MUSIQ/CLIPIQA scores remain nearly unchanged when comparing models with and without the LRT, while the LRT substantially improves pixel-level fidelity. This confirms that RestoreVAR’s gains in fidelity are not achieved at the expense of perceptual realism.
>
> Together, these results demonstrate that RestoreVAR does not simply occupy another point on the perception–distortion plane. Rather, it closes the fidelity gap of LDM-based generative AiOR models while retaining their strong perceptual quality and generalization, and does so with enhanced computational efficiency. This combination of high perceptual quality, improved fidelity, strong robustness, and fast inference supports our claim that RestoreVAR achieves state-of-the-art performance among LDM-based generative AiOR methods.
>
> Key points from this discussion are included in lines 455-469 of the revised paper.

---

### Official Review · Reviewer_DSe5 · 2025-10-26

**Soundness:** 3
**Presentation:** 3
**Contribution:** 3
**Rating:** 6
**Confidence:** 4

**Summary:**

This paper introduces RestoreVAR, an all-in-one image restoration (AiOR) method based on visual autoregression (VAR). The authors improve VAR, originally designed for image generation, to enable it to handle multiple degradation types (dehazing, rain removal, snow removal, deblurring, and low-light enhancement) in a single model. Key technical contributions include: (1) a cross-attention mechanism to inject semantic information from degraded images; (2) a lightweight latent refinement transformer (LRT) to preserve fine details; and (3) a VAE decoder fine-tuned on a continuous latent vector rather than discrete labels. RestoreVAR achieves state-of-the-art performance among LDM-based generative methods, with inference speed 10 times faster than diffusion models, but significantly underperforms non-generative methods on some metrics.

**Strengths:**

1.RestoreVAR is the first work to apply Visual Autoregressive Modeling (VAR) to image restoration tasks in a generative setting, demonstrating the potential of autoregressive models beyond traditional generation tasks.

2.Across five degradation types, RestoreVAR achieves state-of-the-art performance among current LDM-based generative approaches, while also providing a 10× faster inference speed.

3.The ablation studies are well-designed and verify the effectiveness of each component in the proposed framework.

**Weaknesses:**

1.The paper compares RestoreVAR with both non-generative AiOR models and LDM-based generative methods. However, on many standard metrics, non-generative methods still outperform RestoreVAR. Although the paper acknowledges this and attributes the limitation to the VAE, it could further clarify when and why a generative approach is preferable in practice and quantify the trade-offs.

2.The fine-tuning of the VAE decoder on continuous latent variables seems like a heuristic workaround, but the paper does not sufficiently justify the correctness or theoretical soundness of this approach.

3.The user study reports average scores from 36 participants but does not provide confidence intervals or statistical tests. The evaluation lacks statistical significance analysis, confidence reporting, and detailed methodology, which are essential to support claims of perceptual quality superiority over non-generative methods.

4.The paper briefly mentions limitations but does not provide detailed failure case analysis to illustrate when and why the method may fail.

**Questions:**

1.According to the ablation study, continuous latent fine-tuning improves performance by around 20%. Could the authors elaborate on why the original discrete VAR struggles so significantly in restoration tasks?

2.How does the method perform on real-world mixtures of degradations that do not strictly fall into the five predefined categories?

3.What is the impact of varying the value of K (number of scales) on the trade-off between speed and restoration quality?

4.Can the authors provide a rigorous perceptual quality evaluation to justify that the gap on certain quantitative metrics between RestoreVAR and non-generative methods is acceptable in practice?

---

> ### Author Response · Authors · 2025-11-21
> **Response to reviewer DSe5 (W1, W2, W3)**
>
> We thank the reviewer for their valuable feedback. We now address the concerns raised.
>
> **W1: It would be helpful to further clarify when and why a generative approach is preferable in practice and quantify the trade-offs.**
>
> LDM-based generative AiOR methods offer enhanced generalization and perceptual quality compared to non-generative methods. However, existing LDM–based methods suffer from low pixel-level fidelity and slow inference speeds. RestoreVAR addresses these limitations by offering comparable generalization and perceptual quality to existing generative methods while being significantly faster and exhibiting much higher pixel-level fidelity (see Table 2 in revised version). In practice, RestoreVAR is preferable when robustness to unseen, mixed, or real-world degradations, and fast inference speeds are important, since its generative priors and architecture enable better performance at much lower computation loads than LDM-based methods. Non-generative methods may be preferable when perceptual quality is not a high priority and if it is known that the test samples come from a distribution close to their training data.
>
> Key points from this discussion are included in Lines 455-469 of the revised paper.
>
> **W2: VAE decoder fine-tuning on continuous latents seems like a heuristic workaround; the paper does not sufficiently justify the correctness or theoretical soundness of this approach.**
>
> In our framework, fine-tuning the VAE decoder on continuous latent variables is not a heuristic workaround but a necessary component for achieving high pixel-level fidelity. As shown in Fig. 4 and Fig. 20, the output of VAR without the LRT and using the vanilla VQVAE decoder (the ``discrete'' column in Fig. 4 and column (b) in Fig. 20) is restored but lacks pixel-level fidelity with the input, which is important for restoration. Achieving high fidelity is challenging primarily because of information loss during quantization. This requires converting the quantized latent to a continuous one (the un-quantized version) for which we propose the LRT. Once this continuous representation is obtained, the original VAE decoder which is trained only on quantized latents cannot correctly map these continuous latents back to the image space. Therefore, fine-tuning the decoder on continuous latents is *required* to ensure proper reconstruction and is a principled extension of the VQ-VAE framework rather than an ad hoc modification. Furthermore, our fine-tuning approach is consistent with those in prior work such as HART.
>
> **W3: The user study does not provide confidence intervals or statistical tests; evaluation lacks statistical significance analysis, confidence reporting, and detailed methodology, which are essential to support claims of perceptual superiority over non-generative methods.**
>
> We now provide rigorous statistical analysis for both the user study and the quantitative metrics in Table 2. For the user study, we compute paired differences between RestoreVAR and each method across all $36$ responses ($\Delta=\text{RestoreVAR} - \text{Method}$), and report $95$% confidence intervals (CI), paired t-tests, and Wilcoxon signed-rank tests. The results, summarized in Table R1, show that RestoreVAR significantly outperforms all methods: every mean improvement $\Delta$ is positive and the $95$% CIs do not cross zero. Furthermore, the t-tests and Wilcoxon signed-rank tests report $p < 10^{-4}$. These results demonstrate that the advantages of RestoreVAR over non-generative methods are statistically significant across user study responses.
>
> **Table R1: Statistical analysis for user study using 95% confidence intervals (CIs), paired t-test, and Wilcoxon signed-rank test. Δ denotes the mean paired difference between RestoreVAR and each method.**
>
> | Method      | Δ        | 95% CI       | p (t-test)         | p (Wilcoxon)      |
> |-------------|----------|--------------|---------------------|--------------------|
> | AutoDIR     | 0.694    | ± 0.257      | 6.49×$10^{-6}$          | 3.06×$10^{-6}$         |
> | AWRaCLe     | 2.028    | ± 0.245      | 7.15×$10^{-18}$         | 2.91×$10^{-11}$       |
> | PromptIR    | 2.250    | ± 0.221      | 1.12×$10^{-20}$         | 2.91×$10^{-11}$        |
> | DCPT        | 1.944    | ± 0.262      | 2.15×$10^{-16}$         | 2.91×$10^{-11}$       |
> | InstructIR  | 1.417    | ± 0.293      | 3.28×$10^{-11}$        | 1.66×$10^{-6}$         |

---

> ### Author Response · Authors · 2025-11-21
> **Response to reviewer DSe5 (W3 continued, Q1, Q2)**
>
> **Response to W3 continued:**
>
> We additionally perform statistical significance analysis for the quantitative results in Table 2. Specifically, we compute per-image paired differences between RestoreVAR and each non-generative method on MUSIQ and CLIPIQA and apply dataset-weighted paired t-tests to ensure that each dataset contributes equally to the global statistic. As shown in Table R2, RestoreVAR consistently achieves large positive improvements with tight CIs and extremely small $p$ values. Once again, these results confirm that RestoreVAR achieves statistically significant gains in perceptual image-quality metrics over non-generative methods.
>
> **Table R2: Statistical analysis for results from Table 2 using 95% confidence intervals (CIs), paired t-test, and Wilcoxon signed-rank test. Δ denotes the weighted mean paired difference between RestoreVAR and each method.**
>
> | Method     | MUSIQ Δ | MUSIQ 95% CI | MUSIQ p-value        | CLIPIQA Δ | CLIPIQA 95% CI | CLIPIQA p-value      |
> |------------|---------|--------------|------------------------|-----------|------------------|------------------------|
> | AWRaCLe    | 5.084   | ± 0.554      | 2.5×$10^{-55}$            | 0.0536    | ± 0.0080        | 8.5x$10^{-34}$            |
> | DCPT       | 5.206   | ± 0.613      | 3.0×$10^{-49}$           | 0.0166    | ± 0.0079        | 3.9×$10^{-5}$            |
> | DFPIR      | 8.276   | ± 0.757      | 2.5×$10^{-71}$          | 0.0733    | ± 0.0093        | 3.6×$10^{-44}$            |
> | InstructIR | 6.984   | ± 0.902      | 1.3×$10^{-42}$          | 0.0495    | ± 0.0065        | 1.8×$10^{-41}$            |
> | PromptIR   | 7.002   | ± 0.648      | 3.3×$10^{-70}$            | 0.0395    | ± 0.0068        | 8.4×$10^{-27}$            |
>
> We now provide additional details for the user study methodology. We collected 36 independent responses. Participants evaluated each result on a five-point scale for (i) restoration performance, (ii) overall quality, and (iii) scene consistency. The survey was conducted using Qualtrics (Provo, UT). Screenshots of the survey are given in Figs. 15 and 16. Fig. 15 shows the introduction page of the user study. The user would be provided with a randomly sampled google drive link consisting of restored outputs of different methods, where each method is assigned a number. The user needs to evaluate each of these images on the afore-mentioned criteria and enter the scores in the table shown in Fig. 16. Finally, the collected data is analyzed to arrive at the afore-mentioned conclusions.
>
> **W4: No detailed failure case analysis to illustrate when and why the method may fail.**
>
> We found that RestoreVAR does not perform well when the input degradation is extremely severe. Fig. 25 shows the result for a very dark low-light sample from the SID [sid] dataset. The restored output looks inferior to the GT due to the very severe nature of the degradation.
>
> We have included this failure case in Appendix Sec. J of the revised paper.
>
> [sid] Chen, Chen, et al. "Learning to see in the dark." CVPR 2018.
>
>
> **Q1: Continuous latent fine-tuning improves performance by around 20%. Why does the original discrete VAR struggle so significantly in restoration tasks?**
>
> As explained for **W2**, the original discrete VAR does not struggle in removing degradations. For instance, in Fig. 4 and Fig. 20, the output of VAR without the LRT and vanilla VQVAE decoder (the ``discrete'' column in Fig. 4 and column (b) in Fig. 20) exhibits strong mitigation of degradation. But it lacks pixel-level fidelity with the input due to the effect of quantization, which is mitigated by the LRT and VAE decoder.
>
> **Q2:How does the method perform on real-world mixtures of degradations that do not strictly fall into the five predefined categories?**
>
> Table 2, Figs. 6 and 22  show results for TOLED and POLED datasets that contain real-world mixtures of degradations that do not strictly belong to the five training degradations. TOLED images exhibit blur and noise introduced by the transparent OLED layer, while POLED samples exhibit low-light degradation, color imbalance, and noise caused by low transmittance of the Pentile OLED display [udc]. On TOLED, RestoreVAR significantly outperforms existing non-generative approaches with much better restoration performance as illustrated by the quantitative and qualitative results. On POLED, our approach achieves much higher MUSIQ scores and also produces visually sharper results.
>
> [udc] Zhou, Yuqian, et al. "Image restoration for under-display camera." CVPR 2021.

---

> ### Author Response · Authors · 2025-11-21
> **Response to reviewer DSe5 (Q3, Q4)**
>
> **Q3: What is the impact of varying the value of K (number of scales) on the trade-off between speed and restoration quality?**
>
> Since VAR is an autoregressive model pre-trained with residual quantization, subsequent scales can be predicted only when all previous scales are provided. Thus, it is not possible to test using fewer than $K$ scales when the VAR transformer is trained with $K$ scales. Training variants of RestoreVAR with different numbers of scales and analyzing the resulting trade-off between computational cost and restoration quality is an interesting direction for future work.
>
>
> **Q4: Can the authors provide a rigorous perceptual quality evaluation to justify that the gap on certain quantitative metrics between RestoreVAR and non-generative methods is acceptable in practice?**
>
> In Table R3, we provide no-reference metrics (MUSIQ and CLIPIQA scores) which gauge perceptual quality for the test sets in Table 1, where there is a significant gap between non-generative methods and RestoreVAR on pixel-level metrics. From the MUSIQ/CLIPIQA scores provided in Table R3, it can be observed that the perceptual quality of RestoreVAR is on-par or even higher than those of non-generative methods, indicating perceptually comparable results. However, due to lower pixel-level fidelity than non-generative approaches, RestoreVAR scores lower on pixel-level metrics.
>
> We have included this experiment in Appendix Sec. E of the revised paper.
>
> **Table R3: Comparison of perceptual quality of RestoreVAR and non-generative methods on RESIDE, Snow100k, Rain13K, LOLv1, and GoPro datasets using MUSIQ↑ and CLIPIQA↑ metrics.**
>
> | Method | GoPro MUSIQ | GoPro CLIPIQA | LOLv1 MUSIQ | LOLv1 CLIPIQA | RESIDE MUSIQ | RESIDE CLIPIQA | Rain13K MUSIQ | Rain13K CLIPIQA | Snow100K MUSIQ | Snow100K CLIPIQA | Avg MUSIQ | Avg CLIPIQA |
> |--------|-------------|----------------|--------------|----------------|---------------|------------------|----------------|-------------------|------------------|--------------------|-----------|--------------|
> | PromptIR | 35.87 | 0.155 | 50.24 | 0.306 | 66.54 | 0.434 | 64.63 | 0.471 | 63.29 | 0.417 | 56.11 | 0.357 |
> | InstructIR | 53.06 | 0.251 | 67.55 | 0.364 | 67.59 | 0.449 | 66.51 | 0.504 | -- | -- | 63.68 | 0.392 |
> | AWRaCLe | 44.953 | 0.173 | 63.198 | 0.459 | 61.938 | 0.346 | 64.683 | 0.598 | 62.334 | 0.512 | 59.421 | 0.418 |
> | DCPT | 46.12 | 0.180 | 67.73 | 0.404 | 66.46 | 0.432 | 62.63 | 0.437 | -- | -- | 60.73 | 0.363 |
> | DFPIR | 48.03 | 0.190 | 66.77 | 0.409 | 66.55 | 0.423 | 62.28 | 0.432 | -- | -- | 60.91 | 0.364 |
> | RestoreVAR | 55.45 | 0.231 | 71.47 | 0.396 | 66.45 | 0.481 | 63.83 | 0.516 | 63.56 | 0.482 | 64.15 | 0.421 |

---

### Official Review · Reviewer_NtbG · 2025-10-31

**Soundness:** 2
**Presentation:** 2
**Contribution:** 3
**Rating:** 2
**Confidence:** 4

**Summary:**

The paper extends VAR (Tian et al., 2024) for All-in-one Image Restoration where a single model is trained to handle multiple image degradation types. By adopting VAR as the backbone, the proposed method claims to be much faster at inference time when compared to latent diffusion model-based methods. The method incorporates cross attention with the degraded input latents to reduce the risk of hallucination. It also includes a latent refinement transformer (LRT) to further improve the final output quality. The paper describes a series of experiments to benchmark against existing generative and non-generative methods. Further ablations are presented to justify the design choices such as the LRT and use of continuous latents for conditioning and decoding.

**References:**
Tian, K., Jiang, Y., Yuan, Z., Peng, B., & Wang, L. (2024). Visual autoregressive modeling: Scalable image generation via next-scale prediction. *Advances in neural information processing systems*, *37*, 84839-84865.

**Strengths:**

* The paper extends VAR for image restoration tasks, and discusses how restoration tasks can be framed within this ‘next-scale’ paradigm through the scale space analysis (Sec 3.2).
* The proposed method performs better than other generative methods in terms of PSNR, SSIM and LPIPS (Table 1\) and is also significantly faster than these prior generative methods.
* Experimental comparisons across numerous state of the art generative and non generative methods over standard datasets.

**Weaknesses:**

* In terms of PSNR, SSIM and LPIPS, the proposed method is poorer than existing non-generative baselines. The authors suggest that the generative nature of the proposed method provides better generalisation capabilities and attempt to demonstrate this with comparison on other datasets using referenceless IQA metrics. Given the differences in the datasets, methods benchmarked against (generative and non-generative in Table 1 vs only non-generative in Table 2), and metrics, it is hard to place where this proposed method fits into the wider literature. The paper discusses three different dimensions: i) Multi-task restoration performance, ii) generalizability to unseen degradation types, and iii) computational complexity. A concise discussion of the proposed method in comparison to prior generative and non-generative methods along these three dimensions could be useful.
* The paper claims its strength for generalizability from the priors it has learnt from the VAE (line 84). It claims too that it is the first method to train a VAR directly for AiOR (line 147). Can the authors justify this claim when they have to finetune the VAE decoder and add in a latent refiner transformer? How is this different from HART (which is not compared).

**Questions:**

* Compare computational complexity of proposed method against non-generative baselines.
* For the datasets in Table 2 that have clean/GT images (e.g. REVIDE, POLED, TOLED), is it possible to report PSNR, SSIM, LPIPS?
* How do the different scales contribute to the overall restoration? An ablation study in the style of section 3.2 with {restored, restored+coarse\_deg, restored+fine\_deg, deg} would be helpful to validate that restoring the coarse scales contribute most to overall the restoration performance.
* \[Off tangent\]What are the relative contributions of the VAR backbone and LRT to the overall restoration. Could an LRT \+ VAE decoder acting directly on the output of the VAE encoder outputs (continuous or quantized) produce reasonable results? (i.e. $\\hat{f}\_{cont} \= f^{deg}\_{cont} \+ LRM(f^{deg}\_{cont}, 0)$})
* Can the authors clarify the results presented in Table 1\. Is RestoreVar trained specifically for each of the tasks? If so, is the comparison with Table 1 actually fair, as noted in line 374, non task-specific restoration models are not compared. If not, are the compared models trained with the same datasets (RESIDE, Snow100k, Rain13k, LOLv1, GoPro)?
* What is the performance for other generative methods (Diff-plugin, Auto-DIR, PixWizard) on real-world unseen degradations (Table 2)? Why were they excluded?
* For Table 2, RestoreVar is trained over what datasets?
* What about other VAR restoration methods? VarSR and VarFormer – their exclusion from the experiments makes it hard for this reviewer to gauge the contribution of this proposed method.
* The key claim of the paper’s approach in line 214 is that the degradations are separable from the fine details at various scales. Fig. 2’s illustration are images with synthetic degradation added → how far is this claim valid for real world degradations where the degradations from sensor and environment are mixed in a more complicated manner. Can the authors show the same results for the data used in Table 2?

---

> ### Author Response · Authors · 2025-11-21
> **Response to Reviewer NtbG (W1)**
>
> We thank the reviewer for their valuable feedback. We now address the concerns raised.
>
> **W1: Hard to place where this proposed method fits into the wider literature; A concise discussion of the proposed method in comparison to prior generative and non-generative methods along three different dimensions: i) Multi-task restoration performance, ii) generalizability to unseen degradation types, and iii) computational complexity, could be useful.**
>
> We clarify this weakness by addressing the subsequent questions raised by the reviewer. In summary, we arrive at the following conclusions across the three dimensions mentioned by the reviewer:
>
> 1. Multi-task restoration performance: The paper focuses on all-in-one image restoration, i.e., RestoreVAR and all comparisons in the paper are trained to handle multiple restoration tasks within a unified model. Table 1 contains results for multi-task restoration performance on test sets of the training data using reference-based metrics. Table 1 shows that non-generative models achieve better pixel-level fidelity than LDM-based generative approaches (and RestoreVAR). Among the generative approaches, RestoreVAR achieves far superior pixel-level fidelity.
>
> 2. Generalizability to unseen degradation types: Table 2 (which now includes generative approaches) extensively tests the generalization of all methods (for multi-task restoration) on real-world degradations (LHP, REVIDE, TOLED, POLED and LOLBlur), unseen degradations (TOLED and POLED) and mixture of degradations (TOLED, POLED, LOLBlur and CDD) using no-reference metrics that measure restoration performance and perceptual quality. From Table 2 (or Table R3), it is evident that RestoreVAR significantly outperforms non-generative methods, indicating better generalizability and perceptual quality. Furthermore, the performance of RestoreVAR is on par with those of other LDM-based methods.
>
> 3. Computational complexity: RestoreVAR is significantly faster than existing LDM-based approaches (Table 5). We also now provide computational complexity comparisons with non-generative models in Table 6 (or Table R1). It can be observed that RestoreVAR is not far behind many non-generative approaches, despite requiring multiple inference steps and using more parameters. Nevertheless, it is significantly slower than DCPT and InstructIR.
>
> *Conclusion*: Although non-generative models have fast inference speeds and achieve high pixel-level fidelity, their ability to generalize to real-world, unseen, and mixed degradations is quite limited. On the other hand, existing LDM-based generative approaches offer perceptually superior outputs with far superior generalization capabilities, thanks to the strong priors of their backbones. However, they are very slow and suffer from poor pixel-level fidelity. RestoreVAR overcomes these limitations of existing LDM-based AiOR methods as it is significantly faster (even rivaling some existing non-generative methods), exhibits much higher pixel-level fidelity, with little to no compromise on perceptual quality and generalization performance. We hope our work paves the way to faster, more robust and more faithful generative restoration approaches.
>
> We have included key points from this discussion in Lines 456-469 of the revised paper. To better understand how we arrive at these conclusions please see our responses to questions 1,5 and 6.

---

> ### Author Response · Authors · 2025-11-21
> **Response to reviewer NtbG (W2, Q1, Q2)**
>
> **W2: Paper claims strength for generalizability from the priors it has learnt from the VAE (line 84); Also claims it is the first method to train a VAR directly for AiOR (line 147). Can authors justify this claim when VAE decoder fine-tuning and LRT are needed? How is this different from HART (which is not compared).**
>
> We believe the reviewer is referring to the priors (line 84) learned by the VAR transformer (not the VAE). The LRT and the fine-tuned VAE decoder are introduced primarily to recover pixel-level fidelity, which is crucial for the AiOR task.  However, the restoration capability of RestoreVAR, i.e. the ability to remove degradations, is handled mainly by the VAR transformer. This is illustrated in Fig. 4 and Fig. 20: the output of RestoreVAR's VAR transformer without the LRT and with the vanilla VQ-VAE decoder (the “discrete’’ column in Fig. 4 and column (b) in Fig. 20) exhibits mitigation of the degradation. Thus, generalization to real-world, unseen, and non-standard degradation stems from the VAR transformer, not from the LRT or decoder refinement.
>
> *Difference from HART*: HART was proposed for image generation and not image restoration. Nevertheless, we have compared our LRT and VAE decoder with those of HART (see Table 4, Fig. 19, Lines 288-289 and Appendix Sec. H.2). HART uses a diffusion refiner while our LRT is non-generative. For the image restoration task, our LRT signficantly outperforms the HART refiner while being much more efficient (Table 4 and Fig. 20). Furthermore, Fig. 19 and lines 288-289 show that the reconstruction performance of our fine-tuned VAE decoder is substantially superior to that of HART. This is because HART fine-tunes its decoder on both discrete and continuous latents, while our decoder is fine-tuned only on continuous latents (line 280), making it better aligned with the LRT’s continuous latent predictions.
>
> **Q1: Compare computational complexity of proposed method against non-generative baselines.**
>
> We provide a computational complexity comparison of RestoreVAR with non-generative models in Table R1. Non-generative models are generally faster since RestoreVAR, like other generative approaches, requires multiple inference steps. Nevertheless, RestoreVAR maintains competitive inference speed to methods such as PromptIR, AWRaCLe, and DFPIR, even though it contains more parameters and requires a multi-step generation process. While it is slower than methods such as DCPT and InstructIR, the substantially improved generalization and superior perceptual quality offered by RestoreVAR make it a more robust choice for real-world image restoration scenarios. These results are included in Table 6 and Appendix Sec. B of the revised paper.
>
> **Table R1: Comparison of the computational complexity of RestoreVAR with non-generative approaches.**
>
> | **Method** | **Time (s)** | **Params (M)** |
> |-|-|-|
> | PromptIR   | 0.162  | 35.59  |
> | InstructIR | 0.0234 | 15.84  |
> | AWRaCLe    | 0.188  | 186.68 |
> | DCPT       | 0.0238 | 67.88  |
> | DFPIR      | 0.117  | 182.38 |
> | RestoreVAR | 0.201  | 296.95 |
>
> **Q2: For the datasets in Table 2 that have clean/GT images (e.g. REVIDE, POLED, TOLED), is it possible to report PSNR, SSIM, LPIPS?**
>
> In Table R2, we have provided PSNR, SSIM, and LPIPS scores for the REVIDE, POLED and TOLED datasets. However, these reference-based metrics are not reliable for comparing generalization performance on real-world or unseen degradations. As illustrated in Fig. 6 for the REVIDE dataset, RestoreVAR produces a better restored result than competing non-generative approaches, yet all methods obtain quite similar and low PSNR/SSIM/LPIPS scores in Table R2. In contrast, no-reference perceptual metrics better reflect the quality of restoration and are therefore more appropriate for comparing generalization performance. Prior works such as AutoDIR and AWRaCLe also use no-reference perceptual metrics when assessing generalization performance.
>
> **Table R2: Quantitative comparisons of RestoreVAR against state-of-the-art non-generative approaches on real-world unseen degradations with reference-based metrics.**
>
> | **Method**| REVIDE PSNR↑| REVIDE SSIM↑| REVIDE LPIPS↓| TOLED PSNR↑| TOLED SSIM↑| TOLED LPIPS↓| POLED PSNR↑| POLED SSIM↑| POLED LPIPS↓| Avg PSNR↑| Avg SSIM↑| Avg LPIPS↓|
> |-|-|-|-|-|-|-|-|-|-|-|-|-|
> | PromptIR|16.15 | 0.777 | 0.2636 | 24.21 | 0.738 | 0.2760 | 9.30 | 0.340| 0.6854| 16.55 | 0.6183| 0.4083|
> | InstructIR|12.31 | 0.707 | 0.3921 | 21.18 | 0.756 | 0.2390 | 15.25 | 0.344| 0.6245 | 16.25 | 0.6023 | 0.4185|
> | AWRaCLe|15.61 | 0.760 | 0.2758 | 16.11 | 0.614 | 0.3020 | 8.45 | 0.356 | 0.6412 | 13.39 | 0.5767 | 0.4063|
> | DCPT|14.57 | 0.738 | 0.2921 | 15.14 | 0.630 | 0.3233 | 9.06 | 0.364 | 0.6139 | 12.92 | 0.5773 | 0.4098|
> | DFPIR|15.16 | 0.757 | 0.2712 | 11.92 | 0.558 | 0.3376 | 8.93 | 0.351 | 0.6571 | 12.00 | 0.5553 | 0.4220|
> | RestoreVAR | 16.00 | 0.7373 | 0.2715 | 23.54 | 0.7333 | 0.2182 | 10.60 | 0.3697 | 0.5958 | 16.71 | 0.6134 | 0.3618 |

---

> ### Author Response · Authors · 2025-11-21
> **Response to reviewer NtbG (Q3, Q4, Q5, Q6)**
>
> **Q3: An ablation study in the style of section 3.2 with {restored, restored+coarsedeg, restored+finedeg, deg} would be helpful.**
>
> We now provide an analysis similar to the style of Sec. 3.2 in Fig. 9 which compares (i) the degraded image, (ii) the restored image with its coarse scales replaced by those of the degraded input, (iii) the restored image with only fine scales replaced, and (iv) the restored image. Similar to our observations in Sec. 3.2, replacing the coarse-scale components of the restored output reintroduces the degradation, whereas replacing only the finer scales does not. This once again demonstrates that degradations predominantly reside in the coarse-scales. Furthermore, the restored image with fine-scales replaced appears largely clean indicating that RestoreVAR successfully predicts coarse-scale tokens that are mostly free of degradation. We have included this experiment in Lines 508-523 of the revised paper.
>
> **Q4: Could an LRT + VAE decoder acting directly on the output of the VAE encoder outputs (continuous or quantized) produce reasonable results?**
>
> As explained in the response to **W2**, VAR carries out the restoration while the LRT and VAE decoder ensure fidelity. To demonstrate this, we pass the quantized prediction to LRT as $f^\text{deg}\_\text{cont}$ or $f^\text{deg}\_\text{quant}$. However, note that we still have to pass a meaningful last block signal $z$ as the LRT uses the last block as an important guidance to predict the residual required to estimate the continuous latent. Passing zeros in the place of $z$ provides incorrect guidance to the LRT and produces junk output. Thus, we pass $z$ obtained from the VAR transformer for that particular degraded image. Although $z$ is the hidden representation of the restored output, it is interesting to note that the LRT and VAE are unable to produce a restored image (see Fig. 14). Thus, the LRT and VAE decoder require the restored latent from VAR to produce a degradation-free image, once again underscoring the importance of the VAR transformer in the entire framework. We have included these results in Appendix Sec. D.4.
>
> **Q5: Clarification for results in Table 1.**
>
> RestoreVAR (and all other comparisons) are trained for the all-in-one restoration task, i.e. a single set of model weights is used for all degradation types. InstructIR, DCPT, and DFPIR are originally trained on all the datasets in Table 1 except Snow100k. We were unable to retrain these models for Snow100k as InstructIR and DCPT do not have publicly available training codes and we encountered some technical difficulties in training DFPIR. Other models (and RestoreVAR) are trained on all the datasets in Table 1. Additionally, Table 2 provides comparisons with non-generative models on generalization; no snow-related datasets are used there, ensuring a fully fair comparison.
>
> **Q6: Performance for other generative methods on Table 2 datasets? Why were they excluded?**
>
> We have now included the performance of other LDM-based generative methods on real-world, unseen and mixed degradations datasets in Table 2 of the revised paper and Table R3. It can be observed that RestoreVAR achieves competitive performance on these datasets. The slightly lower metrics for RestoreVAR can be explained by the fact that the latent diffusion backbones (Stable Diffusion [sd]) used by the competing generative models are trained on substantially larger datasets, typically hundreds of millions of images, whereas the VAR backbone is trained on only $\sim 1$M ImageNet-1K [imagenet] images. Despite this, RestoreVAR demonstrates comparable generalization performance while offering much faster inference and improved pixel-level fidelity.
>
> We initially did not provide LDM-based generative methods in Table 2 as it is known that they generalize better and produce perceptually superior outputs than non-generative models (which are in-fact a core motivations of our work). Nevertheless, we agree that including these results  provides useful insights into whether diffusion-based approaches have advantages over RestoreVAR in generalization capability.
>
> [sd] Rombach, Robin, et al. "High-resolution image synthesis with latent diffusion models." CVPR 2022.
>
> [imagenet] Deng, Jia, et al. "Imagenet: A large-scale hierarchical image database." CVPR 2009.

---

> ### Author Response · Authors · 2025-11-21
> **Response to reviewer NtbG (Q6 continued, Q7, Q8, Q9)**
>
> **Response to Q6 continued:**
>
> **Table R3: Quantitative comparisons of RestoreVAR against state-of-the-art non-generative and generative approaches on real-world, unseen, and mixed degradations.**
>
> | **Method**|**LHP** MUSIQ↑|LHP CLIPIQA↑|**REVIDE** MUSIQ↑|REVIDE CLIPIQA↑|**TOLED** MUSIQ↑|TOLED CLIPIQA↑|**POLED** MUSIQ↑|POLED CLIPIQA↑|**LOLBlur** MUSIQ↑|LOLBlur CLIPIQA↑|**CDD** MUSIQ↑|CDD CLIPIQA↑| **Average** MUSIQ↑|Average CLIPIQA↑|
> |-|-|-|-|-|-|-|-|-|-|-|-|-|-|-|
> | PromptIR | 56.780 | 0.366 | 61.191 | 0.459 | 43.218 | 0.281 | 34.536 | 0.303 | 33.693 | 0.166 | 65.895 | 0.483 | 49.219 | 0.343 |
> | InstructIR | 58.269 | 0.359 | 63.116 | 0.416 | 44.985 | 0.298 | 23.317 | 0.241 | 40.221 | 0.202 | 65.491 | 0.482 | 49.900 | 0.333 |
> | AWRaCLe | 57.889 | 0.333 | 59.287 | 0.368 | 44.670 | 0.285 | 40.533 | 0.332 | 38.186 | 0.171 | 66.253 | 0.484 | 51.470 | 0.329 |
> | DCPT | 58.044 | 0.372 | 60.011 | 0.446 | 44.062 | 0.314 | 38.138 | 0.345 | 37.393 | 0.175 | 68.440 | 0.544 | 51.681 | 0.366 |
> | DFPIR | 56.483 | 0.330 | 61.009 | 0.450 | 43.820 | 0.276 | 35.668 | 0.289 | 36.277 | 0.163 | 54.408 | 0.349 | 47.611 | 0.310 |
> | DiffPlugin | 57.351 | 0.420 | 63.483 | 0.406 | 46.219 | 0.311 | 35.086 | 0.407 | 40.054 | 0.212 | 68.791 | 0.578 | 51.831 | 0.389 |
> | AutoDir | 58.085 | 0.380 | 63.918 | 0.416 | 54.585 | 0.341 | 48.796 | 0.380 | 46.642 | 0.225 | 68.575 | 0.571 | 56.767 | 0.386 |
> | PixWizard | 58.600 | 0.411 | 68.487 | 0.420 | 45.804 | 0.329 | 44.305 | 0.348 | 50.708 | 0.268 | 68.409 | 0.589 | 56.052 | 0.394 |
> | RestoreVAR | 57.662 | 0.414 | 63.562 | 0.483 | 52.374 | 0.338 | 48.118 | 0.276 | 46.644 | 0.214 | 68.941 | 0.572 | 56.217 | 0.383 |
>
>
> **Q7: For Table 2, RestoreVar is trained over what datasets?**
>
> Same datasets as Table 1 (Lines 352-358). The goal of Table 2 is to test how well RestoreVAR and other methods generalize to real-world, unseen and mixed degradations when trained on the datasets in Table 1.
>
> **Q8: What about comparisons with other VAR restoration methods: VarSR and VarFormer; Their exclusion makes it hard for this reviewer to gauge the contribution of this proposed method.**
>
> As mentioned in the paper (Lines 81-86 and 144-147), VarSR is proposed only for the super-resolution task and VarFormer is a non-generative model guided by VAR features. We leverage the  scale-space findings in Sec. 3.2 and propose architectural innovations to harness VAR directly for the AiOR task. As requested by the reviewer, we now provide comparisons with VarSR (retrained on same datasets as RestoreVAR) in Table R4. It can be observed that RestoreVAR significantly outperforms VarSR across all degradations. We made our best efforts to compare with VarFormer but were unable to run their code due to technical difficulties.
>
> **Table R4: Quantitative comparison of VarSR and RestoreVAR**
>
> | Dataset | VarSR (PSNR/SSIM/LPIPS) | RestoreVAR (PSNR/SSIM/LPIPS) |
> |---------|---------------------------|--------------------------------|
> | RESIDE   | 21.52/0.762/0.239 | 24.67/0.821/0.074 |
> | Snow100k | 20.14/0.648/0.346 | 24.05/0.713/0.156 |
> | Rain13K  | 20.67/0.659/0.344 | 23.97/0.700/0.153 |
> | LOLv1    | 19.09/0.749/0.234 | 21.72/0.782/0.126 |
> | GoPro    | 20.54/0.668/0.312 | 23.96/0.737/0.167 |
>
> **Q9: The key claim of the paper’s approach in line 214 is that the degradations are separable from the fine details at various scales. How far is this valid for real degradations?**
>
> In Fig. 14, we provide an analysis similar to Fig. 2 for real-world degradations using the POLED and REVIDE datasets. The results indicate that the scale-space decomposition of VAR tends to capture real-world degradations in coarse scales while preserving scene-level details in finer scales. This suggests that the scale-space behavior observed in the synthetic examples of Fig. 2 also extends to real-world degradations. We have included this experiment in Appendix Sec. D3.

---

### Official Review · Reviewer_EKiz · 2025-11-01

**Soundness:** 3
**Presentation:** 4
**Contribution:** 2
**Rating:** 6
**Confidence:** 3

**Summary:**

The paper introduces RestoreVAR, a method for image restoration based on Vision Autoregressive Modelling (VAR) framework. Compared to diffusion-based restoration methods, VAR-based image restoration offers rapid inference speed. The paper details the architecture modifications such as cross attention over the latents of degraded image, an additional latent refinement module for predicting the residual after quantization, and a fine-tuning procedure for the VAE decoder. Combining these changes, the paper shows RestoreVAR can obtain superior performance while while achieving over 10x faster inference.

**Strengths:**

The paper provides a nice visualization on on what is encoded in each scale and finds that VAR captures degradations predominantly in coarse scales and scene-level details in fine scales.

The paper argues that generative model offers strong generalization and convincingly demonstrates it with performance on real-world degradation as well as human preference.

Overall, it's a well executed and presented paper.

**Weaknesses:**

Fine-tuning VAE decoder and adding a Latent Refiner Transformer (LRT) are critical to the final performance of the model. In the meantime, these techniques seem to be transferrable to other methods. So it's unclear whether we can attribute the success solely to VAR.

**Questions:**

1. Could the author include some discussions on methods that speeds up inference for diffusion and how it compares with RestoreVAR? For example, consistency models and rectified flow.

2. In section 4.4, the paper provides an ablation on adversarial and pixel level loss. How about SSIM and perceptual loss?

3. In table 4, the paper reports a significant degradation in performance when the refinement network is trained without the last block output. Could you provide an explanation? Do you see degradation if removing the quantized prediction instead?

4. Have the authors tried inference tricks that boost performance for generation, such as classifier-free guidance? It seems like the degraded image can be treated as a conditioning signal?

5. The model seems to be reliant on conditioning signal after the fine-tuning. Does the model still retain any ability to generation image at all? What does $g_i$ (line 261) look like after the training?

---

> ### Author Response · Authors · 2025-11-21
> **Response to reviewer EKiz (W1, Q1, Q2, Q3)**
>
> We thank the reviewer for their valuable feedback. We now address the concerns raised.
>
> **W1: Fine-tuning VAE decoder and adding a Latent Refiner Transformer (LRT) are critical to the final performance; these techniques seem to be transferrable to other methods; So it's unclear whether we can attribute the success solely to VAR.**
>
> In RestoreVAR, the LRT and VAE decoder fine-tuning is for achieving higher pixel-level fidelity which is crucial for image restoration. However, the core restoration task (removal of degradation) is handled by the VAR transformer, not the LRT and VAE decoder. For instance, in Fig. 4 and Fig. 20, the output of VAR without the LRT and using the vanilla VQVAE decoder (the "discrete" column in Fig. 4 and column (b) in Fig. 20) already exhibits mitigation of the degradation. While LRT-style refinement or VAE decoder fine-tuning on continuous latents can in principle be attached to other latent autoregressive backbones  which also use quantization, these components only refine the fidelity of the model predictions.
>
> Furthermore, Table 4 shows that the performance of the LRT reduces significantly without the last block output of VAR, demonstrating that the continuous latent prediction (and in-turn the decoded results) relies heavily on the representations from the VAR transformer. Thus, the VAR transformer is central to the success of the RestoreVAR framework.
>
> **Q1: Include discussions on methods that speeds up inference for diffusion and how it compares with RestoreVAR; For example, consistency models and rectified flow.**
>
>  We have provided comparisons with DDIM acceleration on Diff-Plugin, AutoDIR and PixWizard in Appendix Sec. B (Figs. 10 (a) and (b)). From Fig. 10 (b), as the inference times of these models approach that of RestoreVAR, their PSNR reduces significantly. Even at higher step counts, these diffusion model-based methods lag by over 1 dB highlighting RestoreVAR's advantages in both speed and performance.
>
> Consistency models and rectified-flow approaches are promising recent developments in accelerating diffusion models. However, both require substantial modifications to the training pipeline. Consistency models demand fine-tuning pre-trained diffusion models with a new consistency objective that enables few-step inference. However, using very few steps can cause over-smoothed outputs that are undesirable for image restoration [wang2025target]. Rectified-flow methods learn a deterministic velocity field along a straight-line path between noise and data, using a flow-matching objective. Although these frameworks speed up inference, they still lack the scale-space insights (Sec. 3.2) offered by VAR, which elegantly fits the objective of all-in-one image restoration.
>
> We have included this discussion in Appendix Sec. B of the revised paper.
>
> [wang2025target] Wang, Cunzheng, et al. "Target-driven distillation: Consistency distillation with target timestep selection and decoupled guidance." AAAI 2025.
>
> **Q2: Ablations on SSIM and perceptual loss for VAE decoder fine-tuning.**
>
> We now provide ablations on fine-tuning the VAE decoder without SSIM and perceptual losses. Fig. 12 illustrates that using only L1 loss produces blurry results. Adding the SSIM loss enhances structural details and incorporating the perceptual loss improves sharpness while preserving the structure (w/o Disc column). Incorporating the discriminator enhances sharpness even further. We have discussed these results in Appendix Sec. D.2 of the revised paper.
>
> **Q3: Why is there a significant performance drop when LRT is trained without last-block output? Do you see degradation if removing the quantized prediction instead?**
>
> As mentioned in Lines 494-496, the last-block output of the VAR transformer serves as a pseudo-continuous guidance that enables the LRT to better predict the continuous restored latent. We refer to it as “continuous guidance’’ because this representation corresponds to the final hidden state of the transformer prior to logit projection for discrete codebook sampling. Without this guidance, the LRT needs to predict the continuous latent solely from the quantized tokens, which is fundamentally ill-posed and challenging as quantization is a many-to-one mapping. This explains the significant performance degradation observed when the last block outputs are removed.
>
> Removing the predicted discrete tokens from the LRT input also leads to a drop in performance (PSNR (dB)/SSIM: 23.04/0.760). In this case, the LRT needs to predict the residual to be added to the discrete latent which is more challenging to estimate without the discrete latent.

---

> ### Author Response · Authors · 2025-11-21
> **Response to Reviewer EKiz (Q4, Q5)**
>
> **Q4: Have the authors tried classifier-free guidance?; degraded image can be treated as a conditioning signal?**
>
> We experimented with classifier-free guidance (CFG), by introducing conditioning dropout during training (i.e., randomly zeroing the degraded input so that guidance can be used at inference). Without conditioning dropout, RestoreVAR achieves 24.67/0.821/0.074 (PSNR (dB)/SSIM/LPIPS) on the RESIDE dataset. Introducing conditioning dropout to enable CFG leads to a measurable decline in pixel-level performance: with $\text{cfg}=1.0$, performance drops to 24.27/0.816/0.076; with stronger guidance ($\text{cfg}=5.0$), the metrics remain around the same at 24.23/0.817/0.076. A reason for this could be a loss of adherence to pixel-level correspondence as we are forcing the model to be more creative during training.
>
> **Q5: Does the model still retain any ability to generation image at all? What does  $g_i$ look like after the training?**
>
> After fine-tuning, the model is fully adapted for the restoration task. This adaptation enforces pixel-level alignment with the degraded input, rather than supporting class-conditional generation. As a result, the model cannot generate class-conditioned images after fine-tuning.
>
> After training, $g_i$ has the values listed in Table R1.
>
> **Table R1: Cross-attention gains $g_i$ for each transformer block.**
>
> | Block $i$ | $g_i$ |
> |-------------|---------|
> | 0  |  0.1590 |
> | 1  |  0.1907 |
> | 2  | -0.1194 |
> | 3  |  0.1377 |
> | 4  | -0.1116 |
> | 5  |  0.0984 |
> | 6  |  0.1056 |
> | 7  | -0.1034 |
> | 8  |  0.1276 |
> | 9  | -0.1270 |
> | 10 | -0.1346 |
> | 11 | -0.1934 |
> | 12 |  0.2505 |
> | 13 | -0.2536 |
> | 14 |  0.3193 |
> | 15 | -0.1674 |

---

### Author Response · Authors · 2025-11-22
**Revision Summary**

We thank the reviewers for their valuable comments and suggestions. Based on the feedback, we have incorporated the following changes in the revised version of the paper:

1. Added discussions on consistency and rectified-flow models (Reviewer EKiz). **See Appendix Sec. B.**

2. Added ablations of SSIM and perceptual losses for VAE decoder fine-tuning (Reviewer EKiz). **See Appendix Sec. D.2.**

3. Added discussion on how RestoreVAR compares with existing LDM-based generative and non-generative approaches across various aspects (Reviewers NtbG, DSe5 and HSQV). **See lines 456-469.**

4. Computational complexity comparisons with non-generative methods (Reviewer NtbG). **See Appendix Sec. B.**

5. Additional scale space analysis on outputs of RestoreVAR (Reviewer NtbG). **See Fig. 9 and lines 508-523.**

6. Results with LRT + VAE on degraded image latents. **See Appendix Sec. D.4.**

7. MUSIQ/CLIPIQA scores for generative methods in Table 2 (Reviewers NtbG and HSQV). **See Table 2 and lines 431-449.**

8. Additional scale-space analysis for real-world degraded images (Reviewer NtbG). **See Appendix Sec. D3.**

9. Statistical tests on user study and results in Table 2, and detailed user study methodology (Reviewer DSe5). **See Appendix Sec. E.**

10. Failure case analysis (Reviewer DSe5). **See Appendix Sec. J.**

11. Perceptual quality scores on Table 1 test sets (Reviewer DSe5). **See Appendix Sec. E.**

12. Perceptual quality metrics for RestoreVAR with and without LRT (Reviewer HSQV). **See lines 500-507.**

We have provided our responses to address the concerns raised by the reviewers. We would be happy to clarify any further questions or concerns and look forward to a fruitful discussion phase.

Authors of Submission 12723

---

### Author Response · Authors · 2025-12-03

Dear Area Chairs,

For ease of reviewing and evaluation, we provide a concise summary of the key strengths highlighted by the reviewers, followed by an overview of the concerns raised and where they are addressed in our rebuttal.

R1: Reviewer EKiz

R2: Reviewer NtbG

R3: Reviewer DSe5

R4: Reviewer HSQV

**Key strengths:**

1. Clear and insightful scale-space analysis (Sec. 3.2) that enables the framing of image restoration within the “next-scale” paradigm of VAR (R1, R2, R4).

2. First to apply VAR to image restoration in a generative setting, demonstrating the broader potential of autoregressive models (R3).

3. Strong generalization to real-world degradations, supported by both perceptual evaluations and user studies (R1).

4. State-of-the-art performance among generative methods (PSNR/SSIM/LPIPS) with significantly faster inference (R2, R3).

5. Extensive comparisons across strong generative and non-generative baselines (R2).

6. Intuitive and reasonable design that leverages VAR’s efficient architecture while offering good sample quality (R4).

7. Well-structured manuscript, clear presentation, and well-designed ablations validating each component (R1, R3, R4).


**Key Concerns:**

We list below the primary concerns shared across reviewers. We have taken care to address all these concerns in our rebuttal and in the revised version of the manuscript. The revision details can be found in our global comment titled “Revision Summary”.

In the points below, W refers to “Weakness” and Q refers to “Question”. For example, R1 W1 refers to our response to “Weakness 1” raised by the reviewer R1.

1. Unclear why VAR is responsible for most of the success of RestoreVAR, when major PSNR/SSIM boost comes from LRT and fine-tuned VAE decoder (R1, R2, R3, R4).

The core restoration capability of RestoreVAR comes from the VAR transformer. The LRT and VAE decoder are introduced to only enhance pixel-level fidelity which is also essential for image restoration tasks. Detailed explanations can be found in the following rebuttal points: R1 W1, R2 W2, R2 Q4, R3 Q1, R4 W2, R4 Q2.

2. How is RestoreVAR placed in the broad literature (R2, R3, R4)?

We provide additional experimental results which show that RestoreVAR offers enhanced pixel-level fidelity, significantly faster inference speed compared to latent diffusion model-based image restoration approaches while retaining similar perceptual quality and generalization performance. Compared to non-generative methods, RestoreVAR offers substantially superior generalization and perceptual quality. More details can be found in R2 W1, R2 Q1, R2 Q5, R2 Q6, R3 W1, R3 Q4, R4 W1, R4 Q1.

3. More scale-space analysis experiments (R2, R4).

We provide scale-space analysis of RestoreVAR outputs which show that RestoreVAR predicts coarse scales that are mostly free of degradation, which aligns well with our motivation. Additionally, we show that the scale-space analysis of VAR also holds for real degradations. These experiments further strengthen our motivations in Sec. 3.2. Details can be found in R2 Q3, R2 Q9, R4 Q2.

In response to the remaining questions by the reviewers, we have provided further comparisons, quantitative analyses, ablations, and clarifications, which we summarize below:

1. Additional ablations (R1 Q2, R1 Q3, R2 Q4, R4 W1, R1 Q4 and R1 Q5).

2. Additional comparisons (R2 Q8).

3. Additional quantitative results (R2 Q2, R2 Q6, R3 W3, R3 Q4).

4. Additional clarifications (R1 Q1, R1 Q5, R2 Q5, R2 Q7, R3 W2, R3 W4, R3 Q2, R3 Q3).

We thank the reviewers for their valuable comments and feedback. We also thank the Area Chairs for their time and effort invested in reviewing and evaluating our work.

Authors of Submission 12723

---

### Meta-Review · Area_Chair_tqFg · 2026-01-08

**Summary:**

Main concern across reviewers was attribution: performance gains seem to rely heavily on the LRT + VAE decoder fine-tuning, so it was unclear how much of the success should be credited to VAR itself. A second theme was positioning and evaluation: reviewers wanted clearer comparison/placement vs prior generative + non-generative methods, including perceptual metrics for other generative baselines and a more explicit perception–distortion discussion. Finally, one reviewer flagged missing rigor in the user study (no statistical tests / methodology) and asked for failure-case analysis. Despite these issues, the paper has a coherent motivation (scale-space analysis), strong speed advantages, and (after rebuttal) a more complete empirical story—so the AC leans borderline accept.

**Reviewer Concerns:**

Addressed in rebuttal
(1) Added statistical testing + clearer user-study methodology; also added significance tests for Table 2 metrics.
(2) Added perceptual metrics (MUSIQ/CLIPIQA) for generative methods in the “generalization” table and expanded perception–distortion discussion.
(3) Quantified perceptual impact of LRT (with/without LRT scores are nearly identical).
(4) Added a concrete failure case (very severe degradation).
(5) Added a direct comparison vs VarSR (retrained); clarified VarFormer could not be run due to technical issues.

Still somewhat outstanding
(1) The “how principled is continuous-latent decoder fine-tuning” justification is improved, but may still read heuristic to some.
(2) Lack of a VarFormer comparison remains (authors state they could not run it).

**Reviewer Scores:**

R1 (EKiz): 6 → 6. Already above threshold; rebuttal addresses their diffusion-acceleration/context questions, but the core “credit assignment” concern likely remains.

R2 (NtbG): 2 → 4. Key evaluation gaps were meaningfully addressed (generative baselines added to Table 2; VarSR comparison added), but missing VarFormer and overall positioning skepticism likely keep this reviewer below the bar.

R3 (DSe5): 6 → 6. Their main rigor requests (stats + failure cases + perceptual evaluation) were addressed, but the theoretical/“principledness” worry probably prevents a confident jump.

R4 (HSQV): 4 → 6. The rebuttal directly targets their two biggest weaknesses (missing perceptual comparisons for generative baselines; whether LRT harms perceptual quality).

---

### Decision · Program_Chairs · 2026-01-26

Accept (Poster)